# Aneuploid senescent cells activate NF-κB to promote their immune clearance by NK cells

Ruoxi W Wang[1], Sonia Viganò[2] (ID), Uri Ben-David[3] (ID), Angelika Amon[1,†] & Stefano Santaguida[2,4,*] (ID)

## Abstract

The immune system plays a major role in the protection against cancer. Identifying and characterizing the pathways mediating this immune surveillance are thus critical for understanding how cancer cells are recognized and eliminated. Aneuploidy is a hallmark of cancer, and we previously found that untransformed cells that had undergone senescence due to highly abnormal karyotypes are eliminated by natural killer (NK) cells *in vitro*. However, the mechanisms underlying this process remained elusive. Here, using an *in vitro* NK cell killing system, we show that non-cell-autonomous mechanisms in aneuploid cells predominantly mediate their clearance by NK cells. Our data indicate that in untransformed aneuploid cells, NF-κB signaling upregulation is central to elicit this immune response. Inactivating NF-κB abolishes NK cell-mediated clearance of untransformed aneuploid cells. In cancer cell lines, NF-κB upregulation also correlates with the degree of aneuploidy. However, such upregulation in cancer cells is not sufficient to trigger NK cell-mediated clearance, suggesting that additional mechanisms might be at play during cancer evolution to counteract NF-κB-mediated immunogenicity.

**Keywords** aneuploidy; complex karyotypes; immune clearance; NF-κB; senescence
**Subject Categories** Cell Cycle; Immunology

## Introduction

Aneuploidy is defined as a state in which the chromosome number is not a multiple of the haploid complement (Pfau & Amon, 2012). In all organisms analyzed to date, an unbalanced karyotype has detrimental effects (Pfau & Amon, 2012; Santaguida & Amon, 2015). In yeast, aneuploidy leads to proliferative defects and proteotoxic stress (Torres *et al*, 2010). The impact of aneuploidy on higher eukaryotes is even more severe. Most single autosomal gains and all autosomal losses cause embryonic lethality. Aneuploidies that do survive embryonic development cause significant anatomical and physiological abnormalities (Lindsley *et al*, 1972; Lorke, 1994; Hassold & Hunt, 2001; Roper & Reeves, 2006). The severe impact of aneuploidy on mammalian physiology is also reflected at the cellular level. Trisomic mouse embryonic fibroblasts (MEFs) and aneuploid human cells proliferate more slowly than their euploid counterparts and experience a variety of cellular stresses (Williams *et al*, 2008; Stingele *et al*, 2012; Santaguida *et al*, 2015; Pfau *et al*, 2016). Among these, aneuploidy-induced replication stress has been extensively studied. Upon chromosome mis-segregation, cells exhibit slow replication fork progression rate and increased replication fork stalling during the following S phase. Replication stress triggers genomic instability and drives the evolution of highly abnormal karyotypes (Sheltzer *et al*, 2012; Ohashi *et al*, 2015; Lamm *et al*, 2016; Passerini *et al*, 2016; Santaguida *et al*, 2017).

Although aneuploidy is highly detrimental at both the cellular and organismal levels in untransformed cells, it is a hallmark of cancer, a disease characterized by uncontrolled cell proliferation (Gordon *et al*, 2012). About 90% of solid tumors and 75% of hematopoietic malignancies are characterized by whole chromosome gains and losses (Weaver & Cleveland, 2006). A high degree of aneuploidy is often associated with poor prognosis, immune evasion, and a reduced response to immunotherapy (Ben-David & Amon, 2020). Given the negative effects of aneuploidy on primary cells, it remains unclear how cells with severe genomic imbalances could gain tumorigenic potential. Furthermore, which aneuploidy-associated molecular features alter immune recognition during tumor evolution remains an active field of research.

By inducing high levels of chromosome mis-segregation followed by continuous culturing, we previously generated cells with abnormal complex karyotypes that eventually cease to divide and enter a senescent-like state. We have named such cells arrested with complex karyotypes (ArCK) cells (Santaguida *et al*, 2017; Wang *et al*, 2018). Prior work indicated that ArCK cells upregulate gene expression signatures related to an immune response that render them susceptible to elimination by natural killer (NK) cells *in vitro* (Santaguida *et al*, 2017). However, the molecular and functional bases for this immune recognition of ArCK cells remained unclear.

1 Department of Biology, Massachusetts Institute of Technology, Howard Hughes Medical Institute, Massachusetts Institute of Technology, Cambridge, MA, USA
2 Department of Experimental Oncology at IEO, European Institute of Oncology IRCCS, Milan, Italy
3 Department of Human Molecular Genetics & Biochemistry, Faculty of Medicine, Tel Aviv University, Tel Aviv, Israel
4 Department of Oncology and Hemato-Oncology, University of Milan, Milan, Italy
*Corresponding author. Tel: +39 02 9437 5074; E-mail: stefano.santaguida@ieo.it
†Deceased

Several pathways could be involved in this process. Nuclear factor-kappaB (NF-κB) is induced under several stress conditions to elicit a pro-inflammatory response (Hayden & Ghosh, 2012; Liu *et al*, 2017). In the canonical NF-κB pathway, stress induction causes IκB kinase complex (IKK) to phosphorylate IκB, thereby marking it for proteolytic degradation (Perkins, 2007). As a result of this degradation, RelA-p50 translocates into the nucleus where it activates expression of pro-inflammatory genes. In the non-canonical NF-κB pathway, phosphorylation and cleavage of p100 trigger the nuclear translocation of the RelB-p52 complex to induce a pro-inflammatory response (Perkins, 2007). Recent studies further suggest that cytosolic nucleic acids lead to cGAS/STING activation in senescent cells, which induces an interferon response via JAK-STAT signaling pathway (Glück *et al*, 2017).

Here, we investigate which innate immune pathways contribute to NK cell-mediated elimination of aneuploid cells and show that the NF-κB pathway elicits pro-inflammatory signals in ArCK cells. Inactivating both canonical and non-canonical NF-κB pathways in cells with an unbalanced karyotype prevents NK cell-mediated killing *in vitro*. Furthermore, we find that the NF-κB signature is upregulated in cancer cell lines possessing a higher degree of aneuploidy. However, this activation no longer enhanced NK cell-mediated killing of cancer cells, raising the possibility that aneuploidy-induced immunogenicity might be present only at the early stage of tumorigenesis and aneuploid cancer cells evolve mechanisms to evade immune clearance.

## Results

### An assay to assess elimination of ArCK cells by natural killer (NK) cells *in vitro*

To address the molecular basis for immune recognition of ArCK cells, we established a co-culture system to monitor the interactions between NK cells and ArCK cells. In this setup, we utilized human, untransformed RPE1-hTERT cells in which chromosome segregation errors were forced by inhibiting the function of the spindle assembly checkpoint (SAC; Santaguida *et al*, 2010). To generate ArCK cells, we synchronized RPE1-hTERT cells at the G1/S boundary and released them into the cell cycle in the presence of the SAC kinase Mps1 inhibitor reversine (Fig 1A). We removed the drug once the cells had undergone one round of aberrant mitosis due to SAC

inhibition. 72 h after inducing chromosome mis-segregation, we exposed cells to the spindle poison nocodazole for 12 h, which allowed us to remove dividing cells by mitotic shake-off (Santaguida *et al*, 2017; Wang *et al*, 2018). We repeated the mitotic shake-off 4 more times to ensure the removal of all cycling cells. Cells that remained on the tissue culture plate by the end of this procedure were highly enriched for the ArCK population (Fig 1A; Wang *et al*, 2018). Importantly, such cell cycle arrest was not due to the prolonged nocodazole treatment since euploid control cells were completely removed after two consecutive rounds of shake-offs (Santaguida *et al*, 2017). We co-cultured ArCK cells with an immortalized NK cell line activated by constitutive IL2 expression (NK92-MI; Tam *et al*, 1999) and monitored their interactions by live cell imaging (Fig 1B).

To quantify the degree of NK cell killing of target cells, we first defined a killing event as a target cell that was (i) engaged by one or multiple NK cells, and (ii) lifted from the tissue culture plate (Fig 1B). We chose these criteria because they coincided with target cell membrane permeabilization as judged by the ability of the nucleic acid dye TO-PRO3 to enter a cell (Figs 1B and EV1A). We tracked individual target cells and recorded the time when each of them was killed during the 36-h live cell imaging. If a target cell divided during the time course, we followed only one of the resulting two cells for the remainder of the assay. We then calculated the cumulative cell death for each condition and generated killing curves at hourly resolution. We found that at a ratio of 2.5 NK cells to 1 target cell, ArCK cells were consistently killed twice as effectively as euploid control cells, during a 36-h co-culture experiment (Fig 1C).

Arrested with complex karyotypes cells hardly divided during the 36-hour time lapse employed in our NK cell killing assay whereas euploid control cells continued to divide (Fig 1D). It was thus possible that the difference in NK cell-mediated cytotoxicity toward euploid and ArCK cells was affected by the fact that NK cells became limiting when co-cultured with euploid cells but not aneuploid cells. To test this possibility, we analyzed the effect of changing the NK cell-to-target cell ratio. We found that even at high NK cell-to-target cell ratio (5:1 and 10:1), ArCK cells were still more effectively killed than euploid controls (Fig 1E). We conclude that NK cells were not limited in our assay. We further note that when a cell divided during observation, we followed only one of the two cells after cell division, which corrected for the bias in target cell number. To address the possibility that NK cells became exhausted during the course of the co-culture experiment, we divided the 36-h assay into two time

---

**Figure 1.  ArCK cells are recognized by natural killer (NK) cells *in vitro*.**

A   Schematic representation for the generation of ArCK cells. Time 0 is defined by the estimated onset of Mps1 inhibitor-induced chromosome mis-segregation.

B   Representative images of euploid control or ArCK cells interacting with NK cells. The NK cell-mediated killing was measured at a 2.5:1 (NK cells: target cells) ratio and was recorded by live cell imaging for 36 h at a 30-min interval. TO-PRO-3 (1 μM) was added to the medium at the same time of NK cell addition to measure cell membrane integrity. Phase contrast (top) and TO-PRO-3 signal (bottom) from the same field were presented. Arrowheads indicate ArCK cell death. All images were acquired at the same exposure time and light intensity. Scale bar 20 μM.

C   Measurement of NK cell-mediated killing of ArCK and euploid control cells (Ctrl) at a 2.5:1 (NK cells: target cells) ratio. 50 randomly chosen target cells were followed for 36 h by live cell imaging per condition per replicate. The cumulative cell death was calculated. $n = 3$ biological replicates; mean ± SEM. The statistical significance was determined using nonparametric Kolmogorov–Smirnov test (KS test) as described in the method section; $P < 0.0001$.

D   Measurement of euploid control (Ctrl) and ArCK cell proliferation without NK cells. Live cell imaging of target cells without NK cells was performed using the same condition as described in the method section. For each condition, 50 cells were randomly chosen at the beginning of the movie as the initial population (indicated by the dashline, $n_{initial} = 50$). The cumulative cell number was recorded. Dot plot of individual data points and mean was presented; $n = 2$ biological replicates.

E   NK cell-mediated cytotoxicity across various NK cell-to-target cell ratios. Either euploid control (Ctrl) or ArCK cells were co-cultured with NK cells at the indicated NK cell: target cell ratios. The cumulative killing of target cells was measured. $n = 3$ biological replicates; mean ± SEM. $P < 0.0001$ for all four NK cell: target cell ratios, KS test.

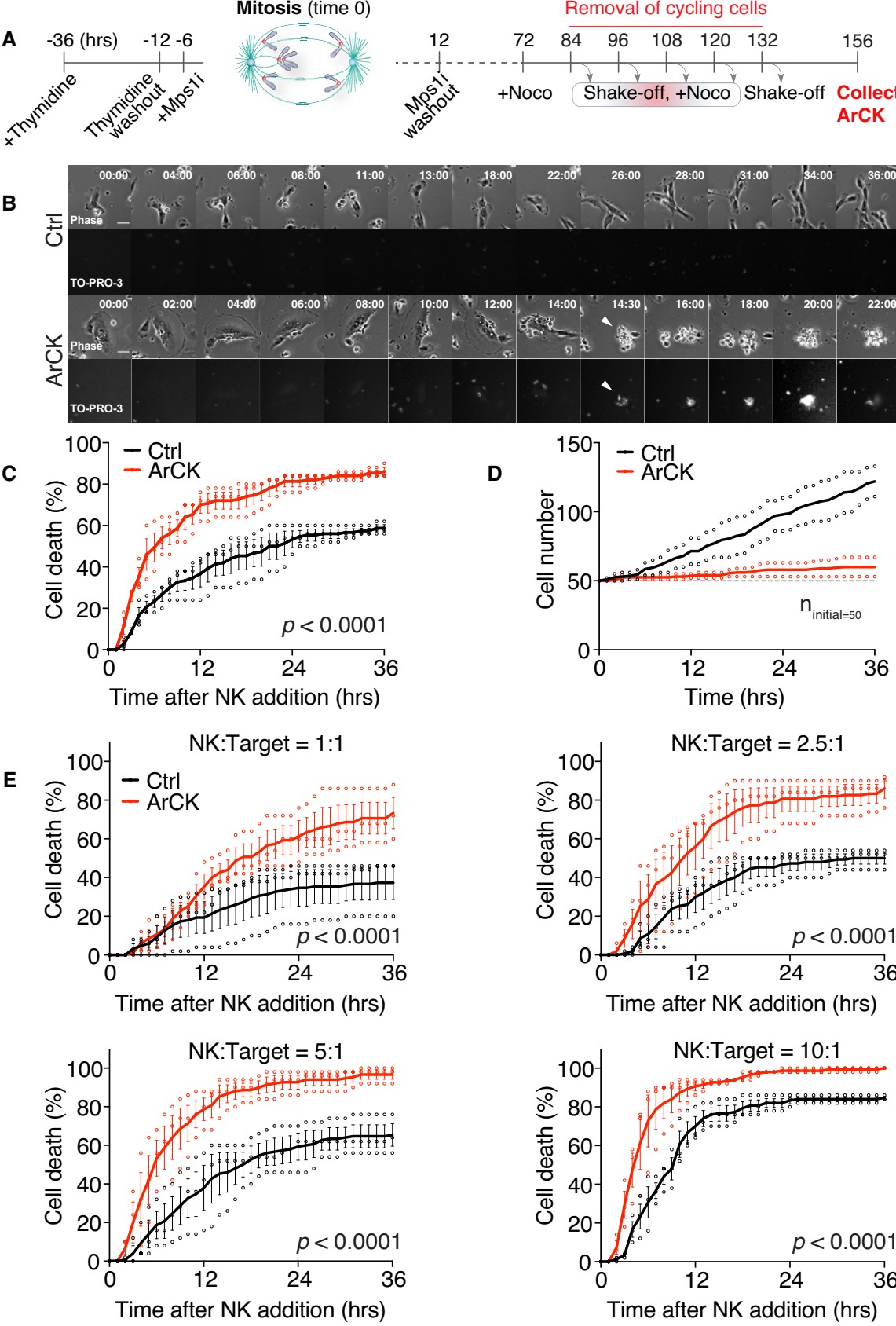

**Figure 1.**

courses, where the same population of NK cells was consecutively co-cultured with target cells for 18 h each. NK cells were equally effective in killing the target cells (Fig EV1B) in this experimental setup, indicating that NK cell exhaustion did not occur within the time course of the analysis. We propose that the eventual plateauing of the killing curve as the assay proceeds is likely due to NK cells taking longer to find their targets.

We next set out to test why euploid control cells are readily killed by NK cells in our *in vitro* assay. One possible explanation was that RPE1-hTERT cells express human telomerase reverse transcriptase (hTERT) and harbor a *KRAS* mutation (Nicolantonio *et al*, 2008), which could generate oncogenic transformation-associated NK cell stimulatory signals (Chiossone *et al*, 2018; Shimasaki *et al*, 2020). To test this possibility, we assessed NK cell-mediated cytotoxicity across three different types of early passage euploid primary fibroblasts derived from normal donors (human embryonic lung fibroblast, IMR90, and normal neonatal or adult human dermal fibroblasts, NHDF-Neo or NHDF-Ad). The analysis of these primary cells revealed large variations in both cell proliferation and NK cell-mediated killing (Fig EV1C and D). Adult human dermal fibroblasts were not readily eliminated by NK cells, whereas both neonatal human dermal fibroblasts and IMR90 cells were highly immunogenic. Thus, it appears that NK cell-mediated killing differs significantly between primary cultured cells. Importantly, we also observed a consistent twofold increase in killing on the Mps1 inhibitor reversine-induced aneuploid NHDF-Ad cells compared with their euploid controls (Fig EV1E), indicating that NK cell-mediated immune clearance of aneuploid cells is not a cell type-specific phenotype. We conclude that in the assay we developed here, highly aneuploid RPE1-hTERT cells are more effectively recognized and eliminated by NK92-MI cells *in vitro* than their euploid counterparts. Since we had developed robust protocols to generate the aneuploid cell population using RPE1-hTERT cells (Santaguida *et al*, 2017; Wang *et al*, 2018), we decided to focus on this cell line to investigate the effects of karyotype alterations on NK cell-mediated immune clearance.

## Prolonged cell cycle arrest associated with features of senescence elicits NK cell-mediated cytotoxicity

Arrested with complex karyotypes cells are largely arrested in G1 and exhibit features of senescence (Santaguida *et al*, 2017; Wang *et al*, 2018). Permanent cell cycle arrest has been shown to elicit an immune response (Gorgoulis *et al*, 2019). To determine whether G1 arrest *per se* is sufficient to cause immune recognition, we assessed NK cell-mediated cytotoxicity toward G1-arrested cells induced by three different methods. We treated RPE1-hTERT cells for 7 days with (i) the topoisomerase II inhibitor, doxorubicin, to induce high levels of DNA damage (Pommier *et al*, 2010); (ii) the cyclin-dependent kinases CDK4/6 inhibitor, palbociclib; or (iii) the imidazoline analog, nutlin3, to disrupt the interaction between p53 and its ubiquitin ligase Mdm2, thereby stabilizing p53. All three conditions have been shown to cause features associated with cellular senescence (Sliwinska *et al*, 2009; Oliveira & Bernards, 2018; Wiley *et al*, 2018).

DNA content analysis by flow cytometry and EdU incorporation showed that after 7 days, all 3 treatments caused the cells to arrest in G1 (Fig 2A–C). With the exception of nutlin3 treatment,

these G1 arrests were irreversible: Most cells did not resume proliferation following drug washout as judged by cell proliferation assays (Fig 2D). Co-culturing these G1-arrested cells with NK cells revealed that irrespective of the means by which the arrest was induced, NK cells exhibited a twofold increase in killing on these G1-arrested cells compared with the untreated proliferating control cells (Fig 2E).

Inactivation of the TORC1 pathway also causes cell cycle arrest (Sousa-Victor *et al*, 2015), but cells enter a quiescent state instead of senescence (Sousa-Victor *et al*, 2015; Kucheryavenko *et al*, 2019). RPE1-hTERT cells were mostly arrested in cell cycle upon treatment with 1 μM of the mTOR kinase inhibitor torin1 after 7 days (Fig 2F). Yet NK cell recognition and killing were not enhanced in cells treated with torin1 (Fig 2G). We conclude that G1 arrest in target cells contributes to NK cell engagement, but only when accompanied by features of senescence.

## Mechanisms triggering senescence contribute to NK cell recognition in ArCK cells

The observation that senescence triggered by multiple mechanisms led to NK cell recognition begged the question of what features in aneuploid cells elicit NK cell-mediated clearance. To address this, we compared a collection of cellular markers contributing to senescence in ArCK cells to those of cells treated with doxorubicin, palbociclib, or nutlin3 for 7 days. First, we assessed DNA damage levels across all conditions by measuring nuclear γ-H2AX foci (Figs 3A and EV2A). DNA damage can increase the expression of NK cell-activating receptor (NKG2D) ligands such as MICA and ULBP2, thereby triggering NK cell-mediated clearance (Raulet & Guerra, 2009). In untreated proliferating control cells, more than 80% of the cells harbored fewer than 10 γ-H2AX foci per nucleus. As expected, doxorubicin caused significantly higher levels of DNA damage in the euploid cells, such that approximately 90% of the cells displayed more than 20 foci and ~50% of this population had 50 foci or more (Fig 3A, panel 2). In contrast, the DNA damage levels in palbociclib- and nutlin3-treated cells were comparable to untreated control cells (Fig 3A, panels 3 and 4). About one third of the ArCK cells harbored more than 10 foci (Fig 3A, panel 5), likely caused by replication stress and/or endogenous reactive oxygen species (ROS) associated with aneuploidy (Li *et al*, 2010; Passerini *et al*, 2016; Santaguida *et al*, 2017).

Induction of the DNA damage response genes p53 and p21 agreed with the presence of γ-H2AX foci with the obvious exception of nutlin3-treated cells (as nutlin3 inhibits Mdm2 to stabilize p53 but does not cause endogenous DNA damage; Fig 3B). We also assessed senescence-associated beta galactosidase activity (SA-beta-gal), a biomarker frequently used to assess degree of senescence. Over 80% of beta-gal-positive cells were observed in doxorubicin-treated and ArCK cells. Palbociclib- or nutlin3-treated cells exhibited a milder increase in the levels of SA-beta-gal-positive cells (~30%), whereas torin1-treated quiescent cells did not show a significant increase in the proportion of SA-beta-gal compared with the untreated control cells (Figs 3C and EV2B). We then examined the senescence-associated secretory phenotype (SASP), which plays a critical role in immune cell recruitment (Gorgoulis *et al*, 2019). We found the composition of SASP varied between different G1 arrests (Figs 3D and EV2C). Nevertheless, arrested cells that did elicit NK

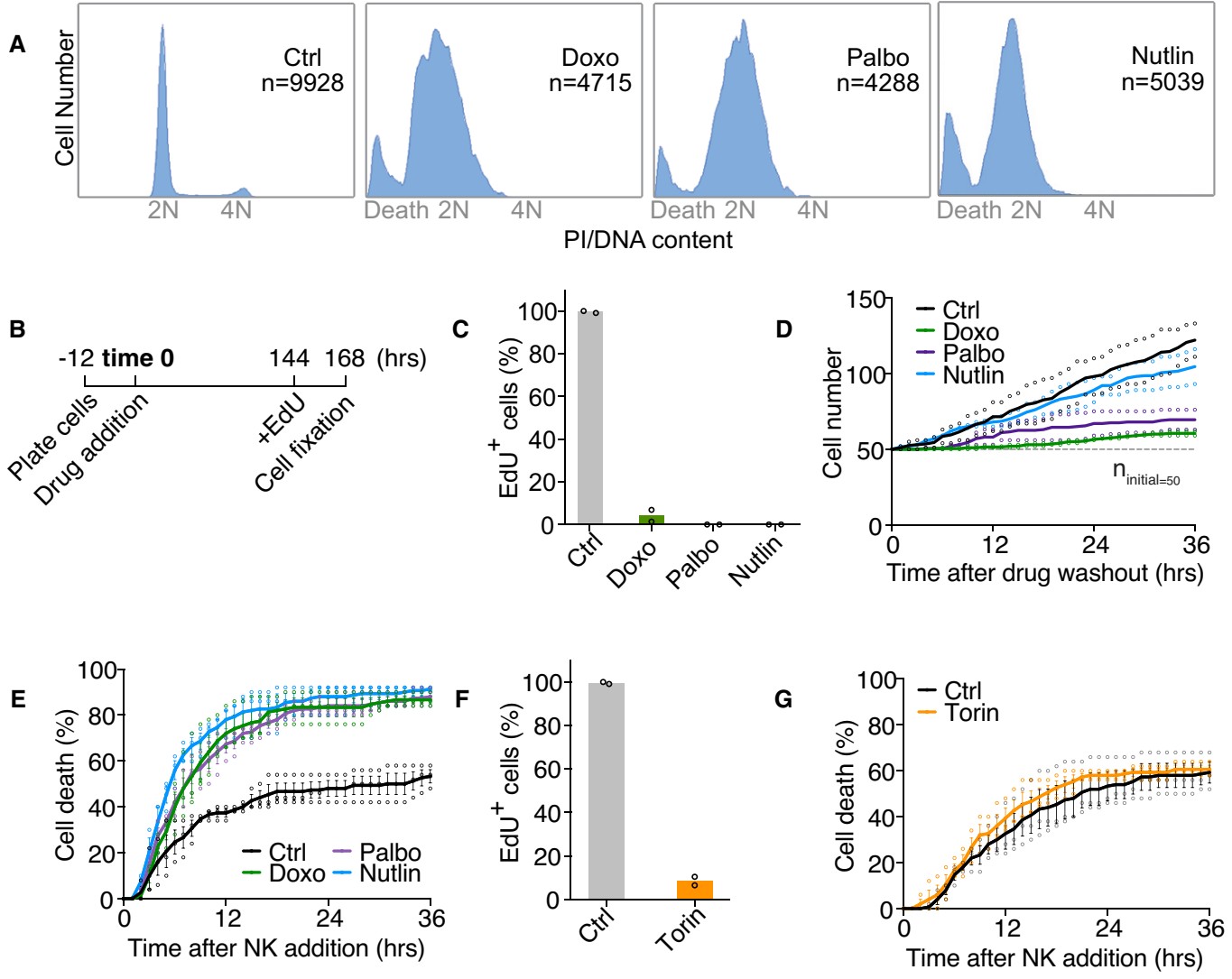

**Figure 2. Prolonged cell cycle arrest associated with features of senescence elicits NK cell-mediated cytotoxicity.**

A   DNA content analysis of various G1 arrests. RPE1-hTERT cells were treated for 7 days with doxorubicin (Doxo; 100 ng/ml), palbociclib (Palbo; 5 μM), or nutlin3 (Nutlin; 10 μM). Total number of cells analyzed is indicated by *n* in each condition. Results were comparable between 2 biological replicates.

B   Schematics of EdU incorporation assay. Drugs were applied to RPE1-hTERT cells 12 h after initial cell plating. 6 days later (144 h), cells were switched to drug medium containing 5-ethynyl-2'-deoxyuridine (EdU; 10 μM) for 24 h before fixation and analysis.

C   The percentage of EdU-positive cells after doxorubicin (Doxo), palbociclib (Palbo), or nutlin3 (Nutlin) treatment. EdU incorporation was performed as described in (B). At least 100 cells were analyzed per condition per replicate. Individual data points and mean are shown; *n* = 2 biological replicates.

D   Cell proliferation (in the absence of NK cells) after 7 days of doxorubicin (Doxo), palbociclib (Palbo), or nutlin3 (Nutlin) treatment. The drugs were washed out after 7 days, and the cells were re-plated for live cell imaging. Cell proliferation was measured as described in Fig 1D. The dashed line indicates the starting cell number ($n_{initial}$ = 50). Dot plot of individual data points and mean are shown; *n* = 2 biological replicates.

E   NK cell-mediated killing for doxorubicin (Doxo), palbociclib (Palbo), and nutlin3 (Nutlin) treated samples (NK cell: target cell = 2.5:1). *n* = 3 biological replicates; mean ± SEM. Ctrl vs. Doxo, *P* < 0.0001; Ctrl vs. Palbo, *P* < 0.0001; Ctrl vs. Nutlin, *P* < 0.0001; KS test.

F   The percentage of EdU-positive cells after 7 days of torin1 treatment was determined as described in (B) and (C). Individual data points and mean are shown; *n* = 2 biological replicates.

G   NK cell-mediated cytotoxicity toward torin1-treated cells. Torin1-treated (Torin) cells were generated as described in (F), and the NK cell killing assay was performed as described in Fig 1. *n* = 3 biological replicates; mean ± SEM. Ctrl vs. Torin, *P* = 0.79, not significant (*n.s.*); KS test.

cell-mediated killing all secreted a plethora of chemokines and cytokines including factors contributing to NK cell recognition [*e.g.*, CCL2 (Robertson, 2002)], whereas the secretion in torin1-treated cells remained low. We conclude that ArCK cells attract NK cells, at least in part, by expressing a canonical senescence immune recognition program.

To determine the role of secreted factors in NK cell-mediated killing on aneuploid cells, we collected the culture medium from

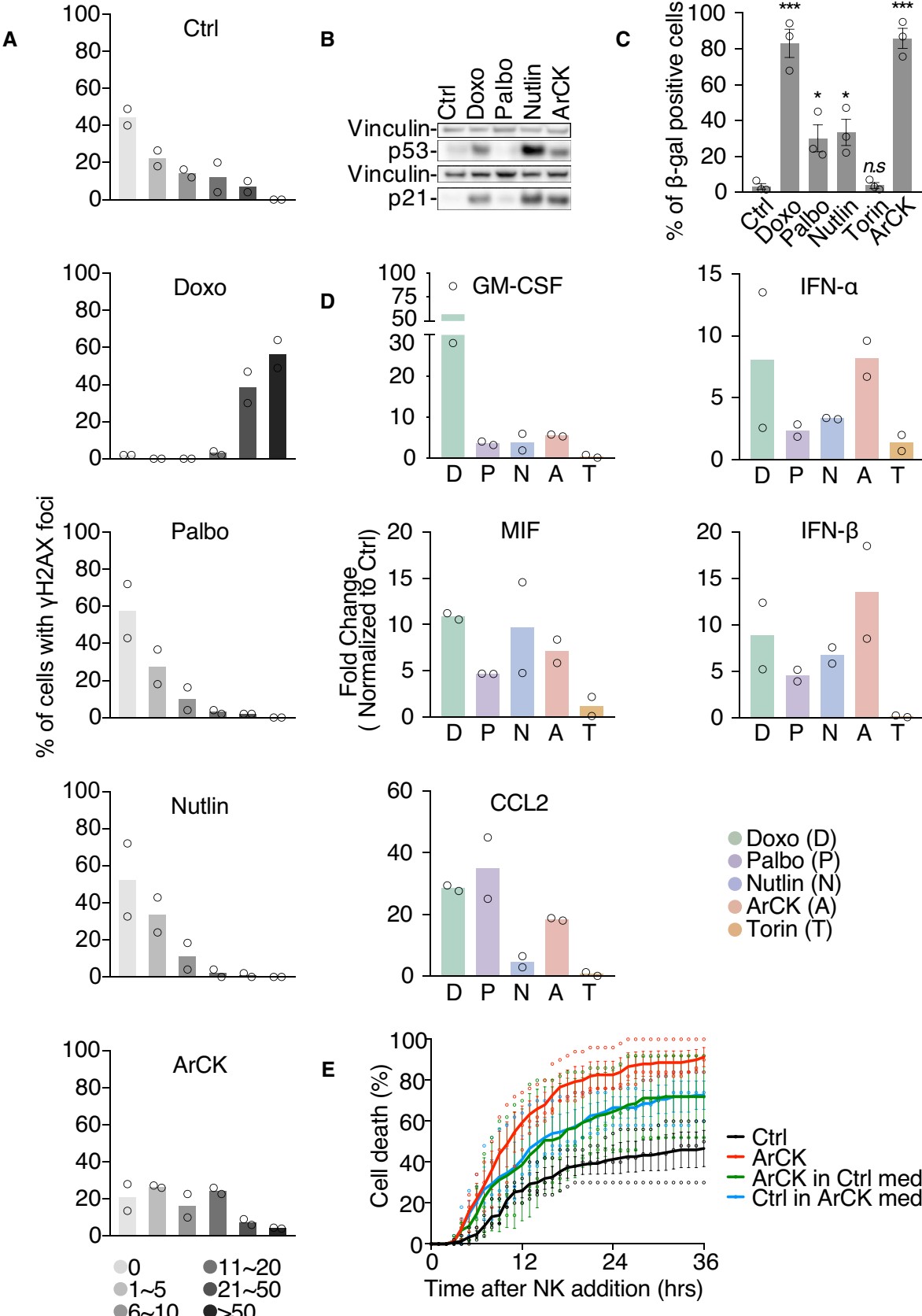

Figure 3.

**Figure 3. Mechanisms triggering senescence contribute to NK cell recognition in ArCK cells.**

A γ-H2AX foci were analyzed in ArCK- and G1-arrested cells (generated as described in Fig 2). At least 50 cells were analyzed per condition per replicate. $n = 2$ biological replicates; individual values and mean are shown. The distribution of the foci number in each treated condition was compared to that of control using Kolmogorov–Smirnov test. Ctrl vs. Doxo, $P < 0.0001$; Ctrl vs. Palbo, $P = 0.07$, n.s.; Ctrl vs. Nutlin, $P = 0.11$, n.s.; Ctrl vs. ArCK, $P = 0.0007$.

B p53 and p21 levels were determined by Western blot analysis. Vinculin was used as loading control. Results were comparable between 2 biological replicates.

C The degree of senescence was measured by senescence-associated β-galactosidase (β-gal) activity. The graph shows the percentage of β-gal-positive cells. At least 100 cells were analyzed per condition per replicate. $n = 3$ biological replicates; mean ± SEM. Ctrl vs. Doxo, ***$P = 0.0006$; Ctrl vs. Palbo, *$P = 0.025$; Ctrl vs. Nutlin, *$P = 0.016$; Ctrl vs. Torin, $P = 0.806$, n.s.; Ctrl vs. ArCK, ***$P = 0.0001$; unpaired $t$-test.

D Secreted cytokine and interferon levels were determined in cell supernatants. Media were collected after 36 h of incubation with cells grown as described in Figs 1 and 2. Cytokine and interferon levels were shown as fold change normalized to euploid control cells. Individual values and mean are shown; $n = 2$ biological replicates.

E NK cell medium was incubated with either euploid control or ArCK cells for 12 h. At the time of NK cell addition, media were switched between ArCK and euploid control cells (Ctrl). NK cell killing was measured as described in Fig 1C. For reference, NK cell killing of ArCK and euploid control cells (Ctrl) without medium switch were performed side by side and plotted on the graph. Black, euploid control cells without medium switch; red, ArCK cells without medium switch; blue, euploid control cells in ArCK cell condition medium; green, ArCK cells in euploid control cell condition medium. $n = 3$ biological replicates; mean ± SEM. Ctrl vs. ArCK, $P < 0.0001$; Ctrl vs. Ctrl in ArCK med, $P < 0.0001$; Ctrl vs. ArCK in Ctrl med, $P < 0.0001$; ArCK vs. ArCK in Ctrl med, $P = 0.0002$; KS test.

aneuploid cells and applied it to euploid control cells (Fig 3E). Medium previously used to culture ArCK cells for 12 h increased NK cell-mediated cytotoxicity toward euploid cells by ~1.5-fold. However, this conditioned medium switch did not completely abolish the differences in NK cell-mediated killing between aneuploid cells and their euploid counter parts. NK cells were still more efficient at killing ArCK cells than euploid cells that have been cultured in pre-conditioned medium from aneuploid cells (Fig 3E). This could be due to the fact that secreted factors accumulated to higher levels during the 36-hour live cell imaging, and/or the possibility that ArCK cell surface features enable their elimination by NK cells. We conclude that conditioned medium provides one or more factors that upregulate NK cell-mediated killing and that aneuploid cells could generate both cell autonomous and non-cell-autonomous signals that render them susceptible to NK cell-mediated cytotoxicity.

### NF-κB and interferon-mediated pathways are upregulated in ArCK cells

Based on our notion that both secreted factors and cell surface features of aneuploid cells contribute to NK cell-mediated killing, we next aimed to determine how NK cell recognition is induced in aneuploid cells. For this, we profiled the gene expression signature of ArCK cells and compared it to those of doxorubicin-, palbociclib-, nutlin3-, and torin1-treated cells by RNA sequencing. Based on gene set enrichment analysis (GSEA), we found there were common significantly upregulated hallmarks shared by G1 arrests that were associated with features of senescence. For example, the p53 pathway is highly upregulated in ArCK-, doxorubicin-, palbociclib-, and nutlin3-treated cells, but is absent in the torin1-treated quiescent cells (Fig 4A). The hallmark gene set "TNFalpha via NF-κB signaling" was among the most upregulated pathways in aneuploid- and doxorubicin-treated samples (Figs 4A and EV2D). ArCK cells also exhibited increased expression of the interferon alpha and gamma response, immune complement, JAK-STAT, and interleukin (IL)-related pathways (Fig 4A and B). In palbociclib- and nutlin3-treated samples, we did observe a mild upregulation of immune-related signatures, but none of them reached significance (*FDR q value* ≤ 0.05; Figs 4A and EV2D). Interestingly, even though NF-κB signaling upregulation was also significant in torin1-treated cells, they were not recognized by NK cells, suggesting NF-κB activation alone

in quiescent cells is not sufficient to cause NK cell-mediated cytotoxicity (Figs 4A and EV2D).

Given the importance of NF-κB pathway in mediating immune recognition, we further characterized this pathway in aneuploid cells. RNA-seq analysis revealed that both RelA and RelB target genes were significantly upregulated in ArCK cells and this was further confirmed by RT–qPCR analysis (Fig 4C and D). This suggested a possible role of both canonical and non-canonical NF-κB pathways. To substantiate NF-κB activation in aneuploid cells, we employed an NF-κB reporter assay in which the expression and secretion of alkaline phosphatase (AP) is controlled by an NF-κB regulatory element (Signorino *et al*, 2014). Using this assay, we observed a threefold increase in AP secretion 96 h post-reversine-induced chromosome mis-segregation (this would allow for the collection of growth medium at a time point reachable from chromosome mis-segregation without cell splitting or the loss of the AP conditioned medium, but also reasonably close to the time of ArCK collection; Fig 4E). Furthermore, we also observed a significant increase in the nuclear translocation of RelA in ArCK cells (Fig EV3A and B), in agreement with previous reports showing NF-κB activation following chromosome mis-segregation (Vasudevan *et al*, 2020). Altogether, these results indicate that NF-κB pathway is indeed upregulated in ArCK cells.

### Both canonical and non-canonical NF-κB pathways are required for NK cell-mediated killing of ArCK cells

What is the relevance of the NF-κB pathway activation in NK cell-mediated elimination of aneuploid cells? To address this question, we generated *RELA* and *RELB* single and double KO cell lines using CRISPR-Cas9 method. In most of the *RELA* or *RELB* single knockout clones, we did not observe a significant decrease in NK cell-mediated cytotoxicity toward ArCK cells (Figs 5A–D and EV3C and D). However, when we knocked out both *RELA* and *RELB*, NK cell-mediated killing in ArCK cells was significantly reduced to a level comparable to the killing of proliferating euploid controls (Figs 5E and F and EV3E). Similar results were observed when ArCK cells were treated with a NF-κB inhibitor BMS-345541 that interferes with both catalytic subunits of IKK [albeit with different binding affinities (Burke *et al*, 2003; Yang *et al*, 2006)] to block NF-κB activation (Figs 5G and EV3F). Importantly, the NF-κB inhibitor

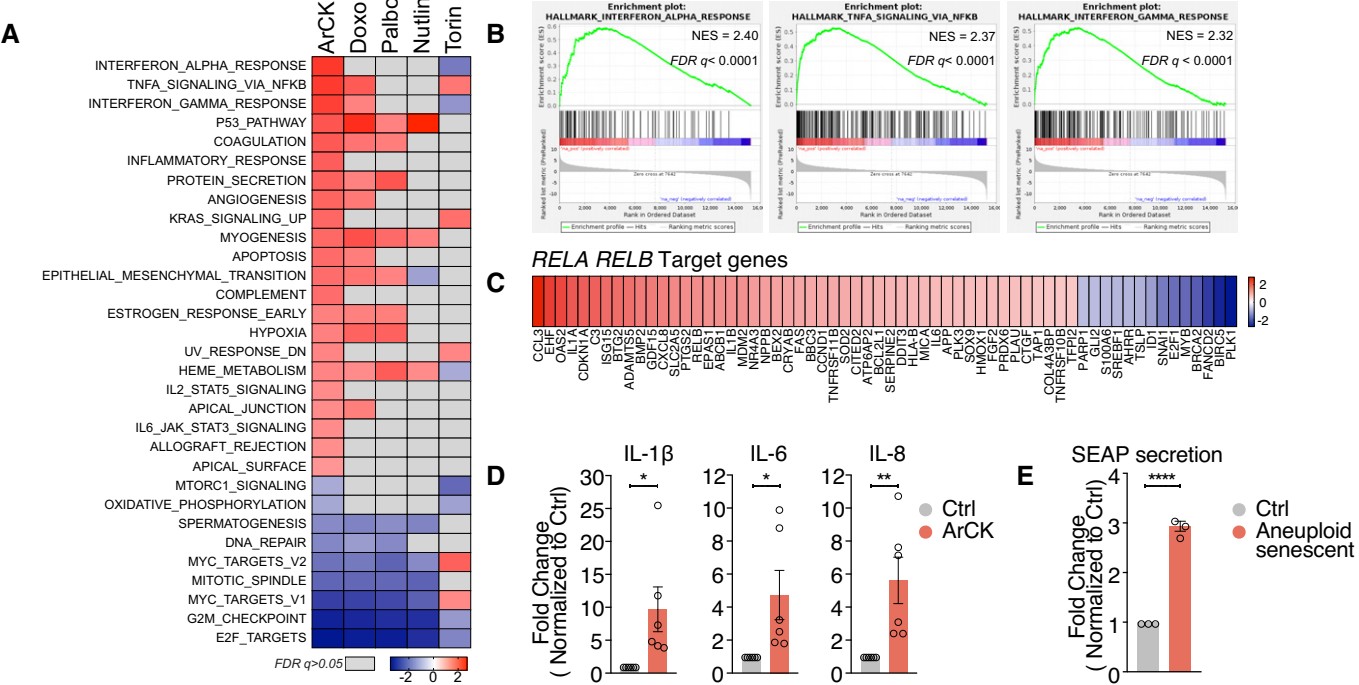

**Figure 4. NF-κB pathway is activated in ArCK cells.**

A   Significantly differentially expressed hallmarks in ArCK cells compared with euploid control cells are shown in the first column in the heatmap (*FDR q* value ≤ 0.05). Normalized enrichment scores are plotted. The corresponding NES for these hallmarks in doxorubicin-treated (Doxo), palbociclib-treated (Palbo), nutlin3-treated (Nutlin), and torin1-treated (Torin) cells is also plotted. Hallmarks did not reach statistical significance (*FDR q* value > 0.05) are shown in gray.

B   Enrichment plots for "Interferon alpha response", "TNFalpha signaling via NF-κB", and "Interferon gamma response" hallmark signatures in ArCK cells compared with the euploid control cells.

C   Significantly differentially expressed *RELA* and *RELB* target genes in ArCK cells compared with euploid control cells were identified by ingenuity pathway analysis based on RNA sequencing data (Log$_2$ fold change, *p*-value ≤ 0.05).

D   RT–qPCR quantifying NF-κB downstream target gene expression in ArCK and euploid control (Ctrl) cells. *n* = 6 biological replicates; mean ± SEM. *IL-1b*, *P = 0.029; IL-6*, *P = 0.032; IL-8*, **P = 0.008; unpaired *t*-test.

E   Measurement of NF-κB activity with NF-κB alkaline phosphatase (SEAP) reporter assay. The reporter was expressed in RPE1-hTERT cells by transient transfection. 10 h after transfection, cells were treated with either DMSO (Ctrl) or reversine (500 nM; Aneuploid senescent) for 96 h and the secretion of alkaline phosphatase in the culture supernatant was measured. The secretion level was normalized to cell number for each condition. *n* = 3 biological replicates; mean ± SEM; unpaired *t*-test, ****P < 0.0001.

treatment in other high immunogenic primary cells (NHDF-Neo) did not lead to a reduction in NK cell-mediated killing (Fig EV3G), suggesting that NF-κB pathway activation in ArCK cells is caused by features associated with aneuploidy induction. We further examined the effect of *RELA RELB* depletion on NK cell-mediated killing in senescent cells triggered by other mechanisms. Based on GSEA analysis, doxorubicin-treated cells also elicit a NF-κB signature (Fig 4A). Indeed, inactivating NF-κB pathway in doxorubicin-treated cells led to a modest but significant decrease in NK cell-mediated killing (Figs 5H and EV3H). We conclude that DNA damage downstream of aneuploidy could be involved, at least in part, in eliciting the immune response in ArCK cells. On the other hand, *RELA* and *RELB* depletion did not rescue NK cell-mediated killing in nutlin3-treated cells, which did not exhibit NF-κB signature (Figs 4A, 5I, and EV3I). This suggests that multiple pathways could be involved in eliciting the immune response, most likely depending on the mechanism leading to cellular senescence. Together, we conclude that aneuploidy triggers NF-κB activation in primary untransformed cells, which contributes to their recognition and elimination by NK cells.

**Retrotransposon activation is involved in triggering immune clearance of ArCK cells**

ArCK cells also induced interferon alpha and gamma responses as judged by RNA-seq and RT–qPCR analysis (Fig EV4A and B). Alpha and gamma interferon responses are primarily mediated by the JAK-STAT pathway (Villarino *et al*, 2017). We confirmed that the activation of these two gene expression signatures was indeed mediated by the JAK-STAT pathway as inactivation of *STAT1* by CRISPR-Cas9 reduced both the interferon alpha and interferon gamma responses in ArCK cells (Fig EV4C and D). To determine the biological relevance of the JAK-STAT response, we depleted *STAT1* in ArCK cells. Our data indicate that deletion of *STAT1* alone is not sufficient to significantly affect NK cell-mediated immune clearance in ArCK cells (Fig EV4E). We speculate that JAK-STAT pathway activation is not essential, but perhaps, potentiates aneuploid cells for immune recognition by NK cells. Remarkably, our data suggest that, although multiple pathways might be activated in aneuploid cells, the NF-κB

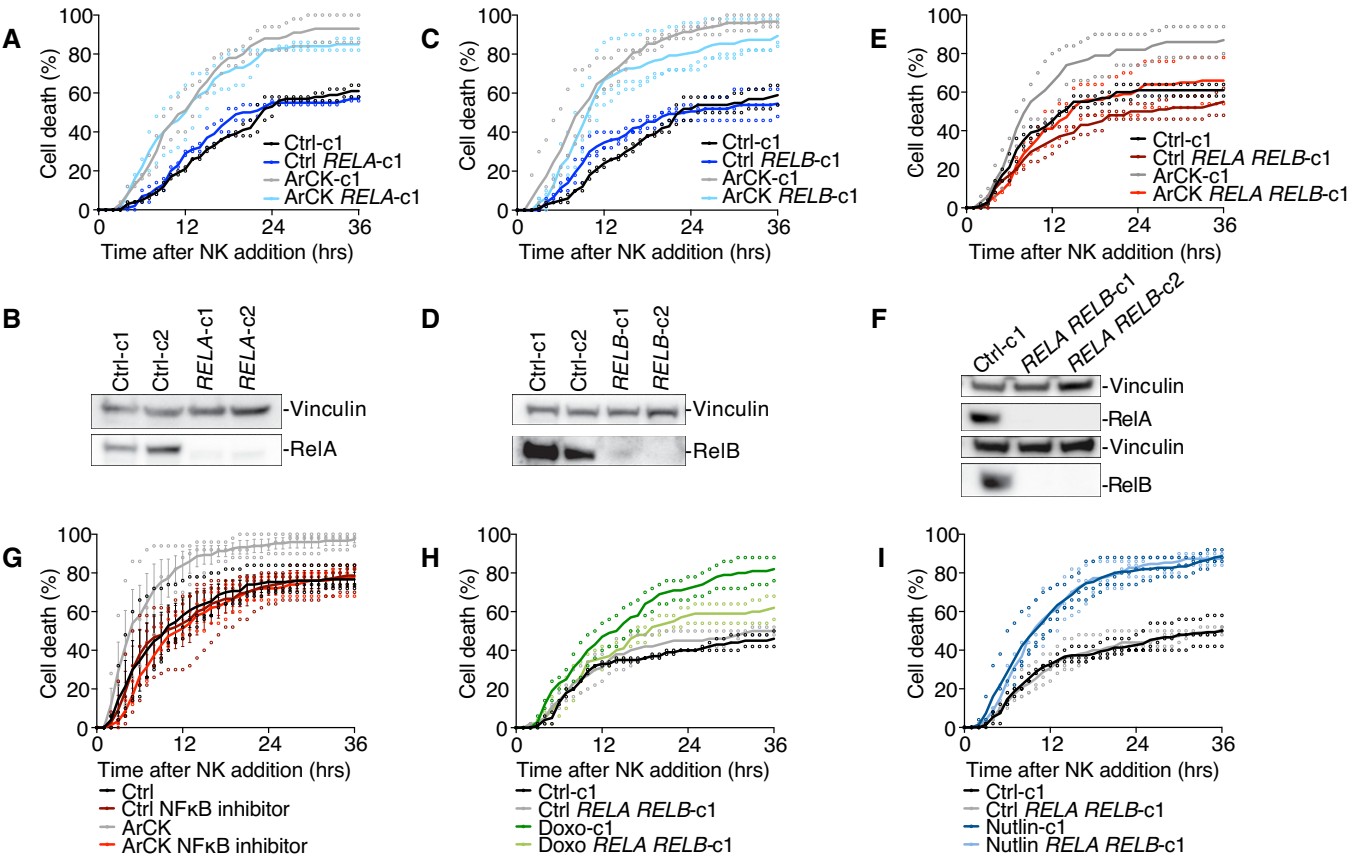

**Figure 5. Both canonical and non-canonical NF-κB pathways are required for NK cell-mediated killing of ArCK cells.**

A   NK cell-mediated killing of *RELA* ArCK cells was compared with clones harboring an empty vector. ArCK *RELA* knockout cells were generated, and the NK cell-mediated cytotoxicity was measured as described in the Method section. Dot plot of individual data points and mean was presented. $n = 2$ biological replicates; ArCK-c1 vs. ArCK *RELA*-c1, $P = 0.70$, n.s.; KS test.

B   Measurement of RelA protein levels in *RELA* KO single cell clones generated in RPE1-hTERT cells.

C   The effect of inactivating *RELB* on NK cell-mediated cytotoxicity in ArCK cells. The same experimental methods were used as described in (A). $n = 3$ biological replicates; ArCK-c1 vs. ArCK *RELB*-c1, $P = 0.47$, n.s.; KS test.

D   Measurement of RelB protein levels in *RELB* KO single cell clones generated in RPE1-hTERT cells.

E   The effect of inactivating both *RELA* and *RELB* on NK cell-mediated cytotoxicity in ArCK cells. $n = 2$ biological replicates; ArCK-c1 vs. ArCK *RELA RELB*- c1, $P = 0.0014$; KS test.

F   Measurement of RelA and RelB protein levels in *RELA RELB* double KO single cell clones generated in RPE1-hTERT cells.

G   ArCK or euploid proliferating control cells were treated with either DMSO or the NF-κB inhibitor BMS-345541 (5 μM) for 48 h before assessing NK cell-mediated cytotoxicity. The drug was washed out during the NK cell co-culture assay. $n = 3$ biological replicates; mean ± SEM. ArCK vs. ArCK NF-κB inhibitor, $P < 0.0001$; KS test.

H, I   The effect of inactivating both *RELA* and *RELB* on NK cell-mediated cytotoxicity in 7-day doxorubicin (H) and nutlin3-treated (I) cells. $n \geq 2$ biological replicates; Doxo-c1 vs. Doxo *RELA RELB*-c1, $P = 0.02$. Nutlin-c1 vs. Nutlin *RELA RELB*-c1, $P = 0.44$, n.s.; KS test.

pathway plays a major role in NK cell-mediated immunogenicity of ArCK cells.

Chromosome mis-segregation also generates micronuclei (Janssen *et al*, 2011; Crasta *et al*, 2012; Liu *et al*, 2018; Martin & Santaguida, 2020). The nuclear envelope of these micronuclei is unstable, leading to their frequent rupture. This in turn causes DNA to spill into the cytoplasm, which activates the cGAS-STING pathway (Dou *et al*, 2017; Harding *et al*, 2017; Mackenzie *et al*, 2017; Bakhoum *et al*, 2018). cGAS-STING could lead to the induction of the NF-κB and interferon response (Dunphy *et al*, 2018). However, in our experimental setup, we did not observe a significant cGAS-STING pathway activation in reversine-induced aneuploid cells as judged by the low degree of IRF3 phosphorylation (Fig EV5A).

Furthermore, *STING* knockout did not affect NK cell-mediated killing in ArCK cells (Fig EV5B). Together, these data suggest that, at least in our *in vitro* assay, NF-κB is the predominant pathway by which aneuploidy induces NK cell-mediated cytotoxicity.

Senescence is also able to induce retrotransposon activation (Cecco *et al*, 2019). This could generate dsRNA intermediates to induce both NF-κB and interferon signatures (Zamanian-Daryoush *et al*, 2000; Alexopoulou *et al*, 2001). Since ArCK cells are senescent (Santaguida *et al*, 2017; Wang *et al*, 2018), we tested whether they might be able to trigger retrotransposon activation. We found that in aneuploid cells, two dsRNA sensors, Rig-I (*DDX58*) and Mda5 (*IFIH1*), were upregulated at the mRNA level compared with their euploid control cells (Fig EV5C). Furthermore, ORF1p, one of the

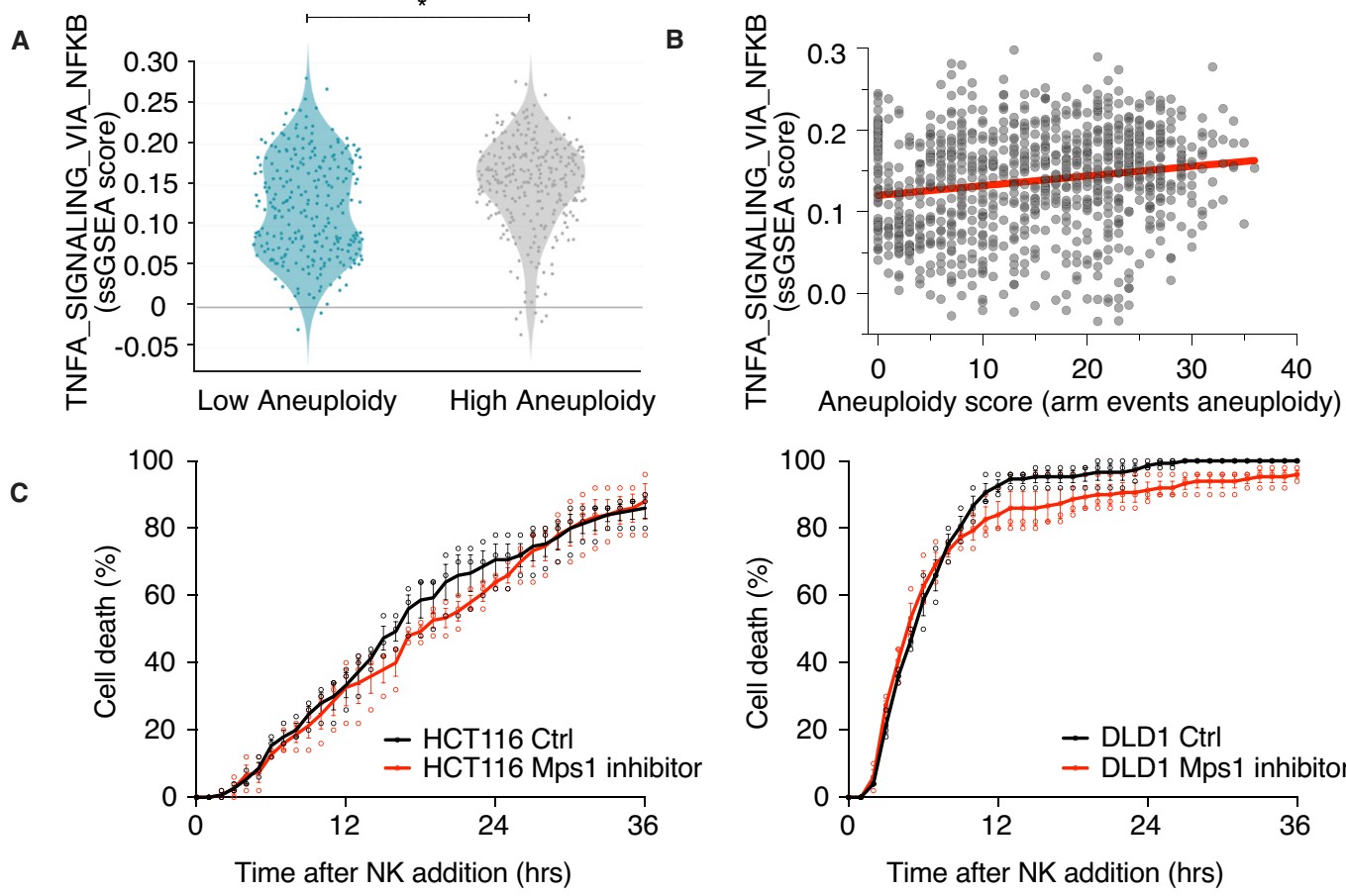

**Figure 6.  NF-κB is active in highly aneuploid cancer cell lines.**

A   Comparison of the Hallmark gene expression signature "TNFA_signaling via NF-κB" in near-diploid (low aneuploidy) and highly aneuploid (high aneuploidy) human cancer cell lines from the CCLE. *P = 5e-08, empirical Bayes-moderated t-statistics. The y-axis represents the ssGSEA expression score. The gray line marks an ssGSEA score of 0, indicating the genes in the hallmark "TNFA_signaling via NF-κB" geneset are not differentially regulated.

B   Correlation plot between ssGSEA expression score of "TNFA_signaling via NF-κB" and the aneuploid score for human cancer cell lines from the CCLE. The trend line for Spearman's correlation plot is indicated in red. Spearman's ρ = 0.181, *P*-value = 2.2e-08.

C   HCT116 (left) or DLD1 cells (right) were treated with either DMSO or the Mps1 inhibitor reversine (500 nM) for 48 h before assessing NK cell-mediated cytotoxicity. The drug was washed out during the NK cell co-culture assay. *n* = 3 biological replicates; mean ± SEM. HCT116 vs. HCT116 Mps1 inhibitor, *P* = 0.44, *n.s.*; DLD1 vs. DLD1 Mps1 inhibitor, *P* = 0.44, *n.s.*; KS test.

proteins encoded by the LINE-1 retrotransposon, was mildly induced in ArCK cells (Fig EV5D). Thus, upregulation of retrotransposons could be relevant to NK cell recognition of aneuploid cells. To suppress retrotransposition, we inhibited reverse transcription by treating cells with the cytosine analog 2',3'-dideoxy-3'-thiacytidine (3TC). We found suppressing retrotransposition led to a partial reduction in NK cell-mediated cytotoxicity toward ArCK cells (Fig EV5E). We conclude that retrotransposon activation in aneuploid cells might be involved, at least partially, in NF-κB upregulation for NK cell-mediated immune clearance in ArCK cells.

### NF-κB pathway is upregulated in highly aneuploid cancer cell lines

Our data indicate that in untransformed cells, aneuploidy induction upregulates NF-κB pathway, which contributes to NK cell-mediated

immune clearance *in vitro*. Interestingly, in tumor cells high levels of aneuploidy correlate with immune evasion (Davoli *et al*, 2017; Taylor *et al*, 2018). It is thus possible that the transformed state of cancer cells suppressed aneuploidy-induced NF-κB signaling. To test this hypothesis, we interrogated the association between NF-κB activation and the degree of aneuploidy in full-blown tumors using the cancer cell line encyclopedia [CCLE; (Barretina *et al*, 2012; Ghandi *et al*, 2019)]. The degree of aneuploidy was scored in almost 1,000 cell lines in the CCLE as recently described (Cohen-Sharir *et al*, 2021). We then created two groups of cell lines, a highly aneuploid and a near-euploid group, defined as the top and bottom quartiles of the number of arm-level chromosome gains and losses, respectively. To assess NF-κB activity in these two cell line groups, we created a ssGSEA signature score (Subramanian *et al*, 2005) for the Hallmark_TNFA_signaling_via_NF-κB gene set and computed the association between this signature and the degree of aneuploidy by

linear regression analysis (see Methods). We found that highly aneuploid cancer cell lines exhibit significantly higher transcriptional signature of NF-κB activity compared with the near-diploid lines (Fig 6A and B). This suggests that aneuploidy could also contribute to NF-κB upregulation in transformed cells. To test whether NF-κB activation in cancer cell lines also contributes to NK cell-mediated killing, we induced aneuploidy in the pseudo-diploid colon cancer cell lines HCT116 and DLD1 using the Mps1 inhibitor reversine. In agreement with previous reports, we observed NF-κB pathway upregulation following chromosome mis-segregation in both cancer cell lines indicated by a significant increase in both RelA and RelB nuclear translocation frequencies (Fig EV5F; Vasudevan *et al*, 2020). However, we did not see an increase in NK cell-mediated killing in reversine-induced aneuploid HCT116 and DLD1 cancer cells (Fig 6C). Our data suggest that although an aneuploidy-associated NF-κB response may still be evident in transformed cell lines, it is not sufficient to enhance the NK cell-mediated immune response.

## Discussion

We previously found that untransformed cells that underwent senescence due to highly abnormal karyotypes are recognized by NK cells *in vitro*. Here, we investigated the molecular mechanism contributing to NK cell-mediated immune clearance and identified NF-κB signaling to be central to the interaction between aneuploid cells and NK cells.

### The NF-κB pathway contributes to the immunogenicity of ArCK cells

Multiple studies have shown that senescent cells are recognized and eliminated by NK cells *in vitro* (Soriani *et al*, 2009, 2014; Iannello *et al*, 2013; Sagiv *et al*, 2016). In this study, we investigated how aneuploidy-induced senescence causes NK cell recognition. We first tested the hypothesis that G1 arrest elicits an NK cell response by comparing NK cell killing kinetics between various G1 arrests. Our analysis revealed that G1 arrest *per se* is not sufficient to cause recognition by NK cells because mTOR inhibition, which caused a quiescence-like G1 arrest, did not elicit an NK cell-mediated killing. Instead, in accordance with other reported NK cell–senescent cell interactions (Iannello *et al*, 2013), we found that the senescence-associated secretory program is primarily responsible for the NK cell-mediated killing of ArCK cells. Medium harvested from aneuploid cell cultures increased the ability of NK cells to kill euploid cells, suggesting that aneuploid cells can establish a pro-inflammatory environment where immune clearance takes place.

Our data revealed that NF-κB pathway upregulation could be one of the major causes of NK cell-mediated immunogenicity of aneuploid cells *in vitro*. We observed upregulation of both the canonical and non-canonical NF-κB pathways in ArCK cells, and inactivation of both pathways, but not of either one of these pathways alone, is sufficient to protect them from NK cell-mediated killing *in vitro*.

Could other immune response inducing pathways contribute to aneuploid cell recognition by NK cells? Chromosome mis-segregation has been linked to induction of an interferon response (Mackenzie *et al*, 2017; Vasudevan *et al*, 2020). Although we

observed an upregulation of the alpha and gamma interferon response in ArCK cells, our data indicate that inactivation of *STAT1*, the major transcription factor mediating the interferon response, did not significantly affect the elimination of ArCK cells by NK cells. Interestingly, *STAT1* activation has been observed in untransformed and cancer cell lines where aneuploidy was induced by continuous exposure to reversine (F. Foijer, personal communication). We speculate that in such experimental setup, persistent DNA damage, which accompanies CIN rather than aneuploidy *per se*, is likely to be the primary cause for JAK/STAT1 pathway activation.

### NF-κB activation in ArCK cells relies on multiple signals

A key question arising from our findings is what causes NF-κB activation in aneuploid cells. Micronuclei—a well-known byproduct of chromosome segregation errors and aneuploidy (Crasta *et al*, 2012) —do not appear to be a major source of immune pathway activation in ArCK cells. On the other hand, while increased retrotransposition could contribute to NF-κB activation in aneuploid cells, the aneuploidy-associated stresses are likely to be the major activators of this immune response. Proteotoxic, oxidative, and genotoxic stresses are defining features associated with the aneuploid state [reviewed in (Santaguida & Amon, 2015)]. We previously found that the surface molecules MICA, MICB, CD155, CD112, ULBP1, and ULBP2—that mediate NK cell recognition—are subtly (about 2-fold) upregulated in ArCK cells (Santaguida *et al*, 2017). MICA and MICB are activated in response to proteotoxic stress, CD112 (also known as Nectin-2) and CD155 (also known as PVR) are expressed in response to DNA damage, and ULBP1 and ULBP2 are the product of both cellular stresses and DNA damage (Raulet & Guerra, 2009). This suggested that multiple features of the aneuploid state contribute to the upregulated immunogenicity. We propose that instead of a unique NK cell-activating feature, the upregulated NK cell-mediated clearance of aneuploid cells is likely mediated by a combination of stresses elicited by the aneuploid state.

### Immune recognition of aneuploidy in cancer

Aneuploidy is a hallmark of cancer that correlates with aggressive disease and immune evasion. Yet in primary cells there is ample evidence that chromosome instability and aneuploidy both elicit a variety of immune responses. For example, micronuclei associated with chromosome mis-segregation activate the interferon response via the cGAS/STING pathway (Harding *et al*, 2017; Mackenzie *et al*, 2017). Chromosome instability upregulates stress-activated protein kinase (SAPK) and c-Jun N-terminal kinase (JNK) pathways, which contribute to inflammatory response (Clemente-Ruiz *et al*, 2016; Benhra *et al*, 2018). MHC complex and antigen processing gene signatures have also been shown to be associated with aneuploidy (Dürrbaum *et al*, 2014). Hence, a crucial question in the field is how malignant transformation dampens aneuploidy and CIN-induced immunogenicity. We investigated whether the aneuploidy-induced NF-κB signature could be down-regulated in highly aneuploid cancer cell lines. Our data argue that this was not the case; indeed, the more aneuploid the cancer cell lines, the higher the NF-κB signaling levels, suggesting that aneuploidy also contributes to NF-κB upregulation in transformed cells. However, we found that the induction of aneuploidy in pseudo-diploid cancer cells did not

further enhance the NK cell-mediated killing. We speculate that in transformed cells, other events occurring during tumorigenesis likely serve to counteract the NF-κB-mediated immunogenicity and render the cancer cells insensitive for NK cell-mediated killing. Interestingly, in a tumor environment, the degree of NF-κB activation inversely correlates with the degree of aneuploidy (Taylor *et al*, 2018), raising the possibility that silencing of the aneuploidy-induced immunogenicity could be a non-cell autonomous event in cancer, perhaps induced by cells in the tumor microenvironment. Understanding which aspects of aneuploidy activate NF-κB signaling and how the activity of the pathway is modulated during tumor evolution will be the critical next steps in understanding the role of aneuploidy during tumorigenesis.

# Materials and Methods

## Cell culture

RPE1-hTERT cells (ATCC Cat# CRL-4000), HCT116 cells (ATCC Cat# CCL-247), and HeLa cells (ATCC Cat# CCL-2) were cultured in Dulbecco's modified Eagle's medium (DMEM, Invitrogen) supplied with 10% FBS (Atlanta Biologicals of South America origin), penicillin/streptomycin (100 U/ml) and L-Glutamine (2 mM). Human primary IMR90 (ATCC Cat# CCL-186), adult, and neonatal normal human dermal fibroblasts (NHDF-Ad and NHDF-Neo; Lonza Cat# CC-2511 and Cat# CC-2509, respectively) were cultured in Eagle's minimum essential medium (EMEM, ATCC) supplied with 10% FBS, penicillin/streptomycin (100 U/ml) and L-Glutamine (2 mM). DLD-1 cells (ATCC Cat# CCL-221) were cultured in RPMI-1640 (Invitrogen) supplied with 10% FBS, penicillin/streptomycin (100 U/ml) and L-Glutamine (2 mM). NK92-MI cells (ATCC Cat# CRL-2408) were cultured in MyeloCult H5100 medium (STEMCELL Technologies). All cells were grown at 37°C with 5% $CO_2$ in a humidified environment.

## Generation of cells arrested with complex karyotypes (ArCK)

To generate ArCK cells, $2.5 \times 10^5$ RPE1-hTERT cells were plated on a 10-cm culture dish and synchronized with thymidine (5 mM) for 24 h. Cells were then released into complete medium. 6 h after thymidine release, cells were switched into medium containing reversine (500 nM). Reversine was washed out 18 h later. 60 h after drug washout, nocodazole (100 ng/ml, Sigma-Aldrich) was added to the culture. 12 h after nocodazole addition, mitotic cells were eliminated from the cell population by shake-off. The shake-off process was performed five times in total at a 12-h interval. The cells left on the plate after five shake-offs were called the ArCK population.

## Generation of cell cycle-arrested cells

To generate G1 arrested samples, $5 \times 10^5$ RPE1-hTERT cells were plated on a 10-cm culture dish and treated with the following drugs: doxorubicin (100 ng/ml, Sigma-Aldrich), palbociclib (5 μM, LC Laboratories), nutlin3 (10 μM, Cayman Chemical), and torin1 (1 μM, LC Laboratories). Cells were collected after 7 days of drug treatment.

## Video microscopy

All live cell imaging was performed using a spinning disk microscope (10× objective) with the environmental chamber maintained at 37°C and 5% $CO_2$ level. Target cells were plated onto 12-well glass-bottom plates in complete normal growth medium at a density $4–6 \times 10^4$ cells/well overnight to allow attachment. Target cells were switched into NK cell growth medium MyeloCult H5100 and incubated for 10 h before starting the live cell imaging. To assess NK cell-mediated cytotoxicity, NK92-MI cells were resuspended into target cell condition medium at the indicated NK cell-to-target cell ratio immediately before the start of the NK cell killing assay. The cell mixture was filmed for 36 h at a 30-min time interval. To assess target cell growth without NK cells, target cells were filmed using the same imaging setting and time scale except no NK cells were added.

## Cell cycle analysis by flow cytometry

Cells were trypsinized and resuspended into 10 ml complete growth medium. After washing twice in cold phosphate-buffered saline (PBS), cells were fixed and permeablized in 70% ethanol at −20°C overnight. Cells were then pelleted and incubated with RNAse (100 μg/ml, Thermo Fisher) and DAPI (1 μg/ml, Thermo Fisher) for 1 h before flow cytometry analysis.

## Immunofluorescence

RPE1-hTERT cells were seeded onto fibronectin (10 μg/ml, Sigma-Aldrich)-coated coverslips at 50-70% confluency and allowed to attach overnight. For RelA and anti-Phospho-histone H2A.X staining, cells were fixed at room temperature with 4% paraformaldehyde in PBS for 15 min and permeabilized with 0.1% Triton X-100 in PBS for 10 min. Cells were then blocked in 3% bovine serum albumin (BSA) in PBS for 40 min. Cells were incubated with primary antibodies for 90 min at room temperature. The RelB staining protocol was adapted from (Vasudevan *et al*, 2020). The following primary antibodies were used as follows: anti-phospho-histone H2A.X (Cell Signaling Technology #9718, 1:500), anti-RelA (Santa Cruz Biotechnology sc-8008 or Abcam ab16502, 1:50), and anti-RelB (Abcam ab180127 or ab33907, 1:1,000). The following secondary antibodies were used as follows: Donkey anti-mouse IgG Cy3 (Euro-Clone, 1:10,000), Alexa Fluor 488 goat anti-Rabbit IgG (Thermo Fisher, 1:1,000). Either Hoechst or DAPI was used to stain DNA. Images were acquired using a DeltaVision (60×) or Leica SP8 Confocal (AOBS, 63× oil objective) microscope. Acquired images were analyzed with Fiji software.

## Western blot analysis

To prepare protein samples, protease inhibitor cocktail (Roche) and phosphatase inhibitor cocktail (Roche) were added to RIPA lysis buffer (Thermo Fisher Scientific) immediately before use. Cells were lysed in cold RIPA buffer, and the lysate concentration was measured by Bradford assay. The lysate was then diluted with loading buffer and heated at 98°C for 5 min. Proteins were resolved on NuPAGE 4-12% Bis-Tris gels (Thermo Fisher Scientific) based on the manufacturer's instructions and transferred onto 0.2-μm PVDF

membranes. Blots were blocked for 1 h at room temperature in OneBlock blocking buffer (Genesee Scientific). Primary antibodies were incubated over night at 4°C. The following primary antibodies were used as follows: anti-GAPDH (Santa Cruz sc-365062, 1:1,000), anti-Vinculin (Sigma-Aldrich V9131, 1:5,000), anti-p53 (Santa Cruz sc-126, 1:200), anti-p21 (Cell Signaling Technology #2947, 1:1,000), anti-p65/RelA (Cell Signaling Technology #8242, 1:1,000), anti-RelB (Abcam #180127, 1:1,000), anti-Stat1 (Cell Signaling Technology #9175, 1:500), anti-IRF3 (Cell Signaling Technology, #11904, 1:1,000), anti-Phospho-IRF3 (Cell Signaling Technology, #4947, 1:1,000), and anti-ORF1p (Cell Signaling Technology #88701, 1:1,000).

### Beta galactosidase staining

ArCK, doxorubicin, palbociclib, nutlin3, torin1, and untreated euploid proliferating RPE1-hTERT cells were plated with normal complete growth medium into 6-well plates at a density of $4 \times 10^5$ cells/well. 24 h later, cells were fixed and stained for the β-Galactosidase activity using a senescence β-Galactosidase staining kit (Cell signaling technology #9860) following manufacturer's instructions.

### Cytokine measurement

ArCK, doxorubicin, palbociclib, nutlin3, torin1, and untreated euploid proliferating RPE1-hTERT cells were generated as described above. 8 ml of complete normal growth medium was placed onto cells and incubated for 36 h. The conditioned medium was harvested, and cell debris was eliminated by centrifugation. To determine the levels of secreted cytokines, condition medium was incubated with proteome profiler human XL cytokine array (R&D Systems, ARY022B) and cytokine levels were measured based on manufacturer's instructions. The levels of interferon alpha and interferon beta in the condition medium were determined using IFN alpha and IFN beta human ELISA kit (Thermo Fisher Scientific, 411001 and 414101, respectively). The total cell number from each sample was measured using a cellometer (Nexcelom). All cytokines, IFN alpha, and IFN beta readings were normalized to cell number.

### RNA sequencing and data analysis

Total RNA was purified using RNeasy Mini Kits (QIAGEN) and sequenced on Illumina HiSeq2000. The RNA-seq data were aligned to a transcriptome derived from the human hg38 primary assembly and an ensembl version 89 annotation with STAR version 2.5.3a (Dobin *et al*, 2013). Gene expression was summarized using RSEM version 1.3.0 (Li & Dewey, 2011) and SAMtools/1.3 (Li et al, 2009). An integer count table for differential expression analysis and $\log_2$ transcripts per million (TPM) with a plus 1 offset for data visualization was prepared with TIBCO Spotfire Analyst (version 7.11.1). Differential expression analysis was done with DESeq2 [version 1.24.0 or version 1.30.0 (Love *et al*, 2014) running under R (version 3.6.0 or version 4.0.3). Pre-ranked Gene Set Enrichment Analysis [versions 2-3.0_beta2, 4.0.3 or 4.1.0, (Subramanian *et al*, 2005)] was run using the DESeq2 Wald statistic as a ranking metric and gene set collections from MsigDB [versions 6.2, 7.0, or 7.2 (Liberzon *et al*, 2015)]. Three biological replicates were included per condition

in the RNA sequencing. The treated samples were compared with the euploid control cells processed and sequenced from the same experiment to avoid batch effect.

### Generation of knockout cell lines using the CRISPR-Cas9 system

Lentiviral CRISPR-Cas9 plasmid LentiCRISPR_v2-Puro (Brett Stringer's Lab) cloned with guide RNA (gRNA) designed by Feng Zhang's lab from the Broad Institute targeting exons of human *RELA* (AGCGCCCCTCGCACTTGTAG), *RELB* (TCGCCGCGTCGCCAGACC GC), and *STAT1* (ATTGATCATCCAGCTGTGAC) was purchased from GenScript. CRISPRv2 constructs along with packaging plasmids pMD2.G (Addgene 12259) and psPAX2 (Addgene 12260) were transfected into 293FT cells (Thermo Fisher, Cat# R70007) using TransIT-LT1 transfection reagent (Mirus). Virus was collected, and the lentiviral titer was estimated by Lenti-X GoStix Plus (TaKaRa). RPE1-hTERT cells were plated at ~60% confluency and infected at a MOI of 1 for *RELA*, *RELB*, or *STAT1* single knockouts, and at a MOI of 2 for *RELA RELB* double knockout. Virus was washed out 20 h post-infection, and the non-infected cells were selected against by puromycin treatment. LentiCRISPR_v2-Puro vector without gRNAs was integrated into RPE1-hTERT cells to generate the control cell line. 2 d after puromycin selection, single cells were sorted into 96 wells and expanded as individual clones. The successful knockout of *RELA*, *RELB*, or *STAT1* was verified by Western blot analysis. All experiments described above were performed in at least two single cell knockout clones.

### RT–qPCR analysis

Total RNA was purified using an RNeasy Mini Kit (QIAGEN). RNA concentration was determined using a NanoDrop. 750 ng of RNA was used for reverse transcription reactions using SuperScript III First-Strand Synthesis SuperMix (Invitrogen). The mRNA levels were then quantified by qPCR using SYBR Premix Ex Taq (TaKaRa) on Roche Light Cycler. Primer sequences: *STAT1*-Forward GATGTT TCATTTGCCACCATCCGTTTTC. *STAT1*-Reverse GGCGTTTTCCAG AATTTTCCTTTCTTCC. *GAPDH*-Forward CCATGTTCGTCATGGGTG TGAACCATG. *GAPDH*-Reverse CCACAGCCTTGGCAGCGCCAGTAG AGG. *DDX58*_Forward CGGCACAGAAGTGTATATTGGATGCATTC. *DDX58*_Reverse GGAAGCACTTGCTACCTCTTGCTCTTC.

*IFIH1*_Forward CTGGGACTAACAGCTTCACCTGGTGTTG. *IFIH1*_Reverse GCATCTGCAATGGCAAACTTCTTGCATG. To measure NF-κB target expressions, 500 ng of RNA was retro-transcribed using OneScript Plus cDNA Synthesis Kit (abm, G236) and the following TaqMan assays were used (Thermo Fisher Scientific): IL-1b, hs00174097_m1; IL-6, Hs00985639_m1; IL-8, hs00174103_m1.

### NF-κB Secreted Alkaline Phosphatase (SEAP) Reporter Assay

A NF-κB Secreted Alkaline Phosphatase Reporter Assay Kit (Novus biologicals, NBP2-25286) was used to measure the secretion of the SEAP protein under the control of the NF-κB promoter. To generate NF-κB secreted alkaline phosphatase reporter cell line, RPE1-hTERT cells were plated in a six-well plate in 2 ml DMEM, containing 10% FBS, L-Glutamine, and nonessential amino acids. 12 h later, cells were transfected with 1 μg pNF-κB /SEAP plasmid using 6 μl Lipofectamine 3000 (Invitrogen, L3000008) per well. 10 h after

transfection, cells were replaced with medium containing either reversine (500 nM) or DMSO for 96 h. The supernatant was then collected to measure the levels of SEAP according to the manufacturer's instructions. The SEAP assay standard curve used to calculate the sample SEAP concentration was generated by loading a serial dilution of SEAP standard on the same plate. The absorbance was measured after 1 h of incubation with the PNPP substrate using a PHERAStar FSX Microplate Reader (BMG LABTECH). SEAP secretion levels were then normalized to cell number for each condition.

### CCLE data analysis

Gene expression data for the CCLE lines were obtained from DepMap 19Q1 DepMap release (www. DepMap.org). A ssGSEA signature (Subramanian *et al*, 2005) score was calculated for the Hallmark_TNFA_signaling_via_NF-κB gene set. Aneuploidy scores were obtained from Ref.(Cohen-Sharir *et al*.). The association between the signature score and the aneuploidy groups was assessed by linear regression analysis, using the R package limma (Ritchie *et al*, 2015). Significance was calculated by empirical Bayes-moderated t-statistics.

### Statistical analysis

To test statistical significance on the killing assay, the raw data corresponding to the time points when a target cell was killed by NK cells were pooled from individual biological replicates for each condition. The cells that were not killed at the end of the 36-h killing assay were assigned to 36 h as the killing time. The distribution of the killing time in each condition was compared, and the significance was determined using nonparametric Kolmogorov–Smirnov test (KS test). Significance was called at $P < 0.05$. All statistical analysis was performed using GraphPad Prism or R software. Details of the statistical tests on data besides the killing assay were stated in the associated figure legends.

## Data availability

The RNA-seq data sets generated for this study can be accessed at Gene Expression Omnibus (GEO) database with the accession number GEO: GSE154919 (http://www.ncbi.nlm.nih.gov/query/acc.cgi?acc=GSE154919).

**Expanded View** for this article is available online.

## Acknowledgements
We thank Floris Foijer for insightful discussions and for sharing data prior to publication. We thank Jacqueline Lees, Iain Cheeseman, Federica Facciotti, and members of the Amon and Santaguida labs for their helpful comments and discussions regarding this project and manuscript. We thank Charlie Whittaker, Dikshant Pradhan, and the Swanson Biotechnology Center for help with the gene expression analysis. This work was supported by grants to S. S. from the Italian Association for Cancer Research (MFAG 2018—ID. 21665 project), Fondazione Cariplo, Ricerca Finalizzata (GR-2018-12367077), the Rita-Levi Montalcini program from MIUR and by the Italian Ministry of Health with Ricerca Corrente and 5 × 1,000 funds and by NIH grant CA206157 to A.A., who was an investigator of the Howard Hughes Medical Institute, the Paul F. Glenn Center for Biology of Aging Research at MIT and the Ludwig Center at MIT. Work in the Ben-David lab is supported by the Azrieli Foundation, the Richard Eimert Research Fund on Solid Tumors, the Tel Aviv University Cancer Biology Research Center, and the Israel Cancer Association. This work is dedicated to the memory of Angelika Amon.

## Author contributions
RWW, AA, and SS conceptualized the study; RWW, SV, UB-D, and SS investigated the study; RWW, UB-D, AA, and SS wrote the manuscript; UB-D, AA, and SS contributed to funding acquisition and supervised the study. All authors discussed the results and commented on the manuscript.

## Conflict of interest
The authors declare that they have no conflict of interest.

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
