## [Review Process File · EMBO Reports]

Aneuploid senescent cells activate NF- κ B to promote their immune clearance by NK cells

Ruoxi Wang, Sonia Viganò, Uri Ben-David, Angelika Amon, and Stefano Santaguida

DOI: [10.15252/embr.202052032](https://doi.org/10.15252/embr.202052032)

Corresponding author(s): Stefano Santaguida (stefano.santaguida@ieo.it)

Review Timeline:

Submission Date:	5th Nov 20
Editorial Decision:	6th Nov 20
Revision Received:	19th Mar 21
Editorial Decision:	22nd Apr 21
Revision Received:	25th Apr 21
Accepted:	19th May 21

Editor: Deniz Senyilmaz Tiebe

Transaction Report: This manuscript was transferred to EMBO reports following peer review at The EMBO Journal.

Dear Stefano,

Thank you for submitting your manuscript to EMBO Reports, which was previously reviewed at another journal. You transferred your manuscript along with the referee reports and your revision plan.

I can see that the proposed revision will significantly strengthen the manuscript. As discussed before, the concerns regarding the clearance of cycling aneuploid cells do not need to be addressed. However, the requirement of NF- κ B signalling for the process need to be strengthened as proposed, and the rest of the specific concerns need to be addressed (including the minor referee concerns).

Having looked at everything, I would like to invite you to revise your manuscript. Please address the referee concerns in a complete point-by-point response. Acceptance of the manuscript will depend on a positive outcome of a second round of review. It is EMBO reports policy to allow a single round of revision only and acceptance or rejection of the manuscript will therefore depend on the completeness of your responses included in the next, final version of the manuscript.

*** Temporary update to EMBO Press scooping protection policy:

We are aware that many laboratories cannot function at full efficiency during the current COVID-19/SARS-CoV-2 pandemic and have therefore extended our 'scooping protection policy' to cover the period required for a full revision to address the experimental issues highlighted in the editorial decision letter. Please contact the scientific editor handling your manuscript to discuss a revision plan should you need additional time, and also if you see a paper with related content published elsewhere.***

1. A data availability section providing access to data deposited in public databases is missing (where applicable).
2. Your manuscript contains statistics and error bars based on $n=2$. Please use scatter plots in these cases.

Supplementary/additional data: The Expanded View format, which will be displayed in the main HTML of the paper in a collapsible format, has replaced the Supplementary information. You can submit up to 5 images as Expanded View. Please follow the nomenclature Figure EV1, Figure EV2 etc. The figure legend for these should be included in the main manuscript document file in a section called Expanded View Figure Legends after the main Figure Legends section. Additional Supplementary material should be supplied as a single pdf labeled Appendix. The Appendix includes a table of content on the first page with page numbers, all figures and their legends. Please follow the nomenclature Appendix Figure Sx throughout the text and also label the figures according to

this nomenclature. For more details please refer to our guide to authors.

Please note that for all articles published beginning 1 July 2020, the EMBO Reports reference style will change to the Harvard style for all article types. Details and examples are provided at <https://www.embopress.org/page/journal/14693178/authorguide#referencesformat>

2) individual production quality figure files as .eps, .tif, .jpg (one file per figure).

3) a .docx formatted letter INCLUDING the reviewers' reports and your detailed point-by-point responses to their comments. As part of the EMBO Press transparent editorial process, the point-by-point response is part of the Review Process File (RPF), which will be published alongside your paper. For more details on our Transparent Editorial Process, please visit our website: <https://www.embopress.org/page/journal/14693178/authorguide#transparentprocess>
You are able to opt out of this by letting the editorial office know (emboreports@embo.org). If you do opt out, the Review Process File link will point to the following statement: "No Review Process File is available with this article, as the authors have chosen not to make the review process public in this case."

4) a complete author checklist, which you can download from our author guidelines (). Please insert information in the checklist that is also reflected in the manuscript. The completed author checklist will also be part of the RPF.

5) Please note that all corresponding authors are required to supply an ORCID ID for their name upon submission of a revised manuscript (). Please find instructions on how to link your ORCID ID to your account in our manuscript tracking system in our Author guidelines ().

6) We replaced Supplementary Information with Expanded View (EV) Figures and Tables that are collapsible/expandable online. A maximum of 5 EV Figures can be typeset. EV Figures should be cited as 'Figure EV1, Figure EV2' etc... in the text and their respective legends should be included in the main text after the legends of regular figures.

7) We would also encourage you to include the source data for figure panels that show essential data.

Numerical data should be provided as individual .xls or .csv files (including a tab describing the data). For blots or microscopy, uncropped images should be submitted (using a zip archive if multiple images need to be supplied for one panel). Additional information on source data and instruction on how to label the files are available .

8) Our journal encourages inclusion of *data citations in the reference list* to directly cite datasets that were re-used and obtained from public databases. Data citations in the article text are distinct from normal bibliographical citations and should directly link to the database records from which the data can be accessed. In the main text, data citations are formatted as follows: "Data ref: Smith et al, 2001" or "Data ref: NCBI Sequence Read Archive PRJNA342805, 2017". In the Reference list, data citations must be labeled with "[DATASET]". A data reference must provide the database name, accession number/identifiers and a resolvable link to the landing page from which the data can be accessed at the end of the reference. Further instructions are available at .

9) Please make sure to include a Data Availability Section before submitting your revision - if it is not applicable, make a statement that no data were deposited in a public database. Primary datasets (and computer code, where appropriate) produced in this study need to be deposited in an appropriate public database (see).

The accession numbers and database should be listed in a formal "Data Availability " section (placed after Materials & Method) that follows the model below. Please note that the Data Availability Section is restricted to new primary data that are part of this study.

Data availability

10) Regarding data quantification, please ensure to specify the name of the statistical test used to generate error bars and P values, the number (n) of independent experiments underlying each data point (not replicate measures of one sample), and the test used to calculate p-values in each figure legend. Discussion of statistical methodology can be reported in the materials and methods section, but figure legends should contain a basic description of n, P and the test applied. Please note that error bars and statistical comparisons may only be applied to data obtained from at least three independent biological replicates. Please also include scale bars in all microscopy images.

I look forward to seeing a revised version of your manuscript when it is ready. Please let me know if you have questions or comments regarding the revision.

Kind regards,

Deniz

Deniz Senyilmaz Tiebe, PhD
Editor
EMBO Reports

We are glad that all three reviewers thought our paper was interesting. All reviewers proposed many critical and insightful experiments for the paper revision. We absolutely agree with their suggestions and have included many new experiments into the revised manuscript per reviewers' comments.

At the same time, the reviewers raised major concerns on whether and how the "aneuploid cycling cells" are also cleared by NK cells. We recognize that a fraction of arrested aneuploid population is presented in the aneuploid cycling sample, yet it will be challenging to separate out a pure aneuploid cycling population using our current experimental system. We appreciate the reviewer for suggestions on potential experiments to address this question. We agree this is an important issue, but, unfortunately, we are not in a position to address this point properly. Consequently, in consultation with the editor, we then removed the data referring to aneuploid cycling cells and made substantial changes in both the text and the figures for justification. We have now moved our submission from *EMBO Journal* to *EMBO Reports* for consideration. We are now presenting in the manuscript a major discovery, namely NF- κ B upregulation triggers NK cell-mediated immune clearance of aneuploid cells, which is, as pointed by the reviewers, a relevant finding. We have addressed all the concerns raised by the reviewers in a point-by-point manner below.

Referee #1:

Review comments for "Aneuploid cells activate NF- κ B to promote their immune clearance by NK cells" by Wang et al. (EMBO Journal)

The immune system plays an important part in the prevention of cancer, via recognition and elimination of cancer cells. However, cancer cells can also develop strategies to evade clearance by this immune surveillance. Therefore, studying the mechanisms of this protective surveillance is highly relevant. It has previously been shown by the same authors that cells with very complex karyotypes undergo senescence and are recognized and eliminated by Natural Killer (NK) cells. In this manuscript, Wang et al. study the underlying mechanisms that give rise to this immune response to these aneuploid senescent cells, referred to as Arrested with Complex Karyotypes (ArCK) cells. Furthermore, they also include aneuploid cycling cells in their study and find that these are also cleared by NK cells, although less effectively. They show that ArCK cells are likely eliminated due to the fact that they display a prolonged G1 arrest and show features of senescence. Via evaluating gene expression levels, they find the NF- κ B pathway to be upregulated in both ArCK and aneuploid cycling cells and they speculate that this could be involved in NK cell mediated clearance. Using knockout cell lines of both the canonical and non-canonical NF- κ B pathway, they discover that only double knockouts lead to a significant decrease in NK cell mediated killing of ArCK and aneuploid cycling cells.

The finding that the NF- κ B pathway plays a central role in clearance of aneuploid cells by the immune system is interesting and would be a relevant finding to the aneuploidy field. However, in this study the most convincing findings are mostly based on ArCK cells, that display many features of senescence and the finding that they are eliminated by NK cells is not novel. The

findings on the cycling aneuploid cells are much more relevant as they are a better model for aneuploid cancer cells. However, the data regarding the clearance of cycling aneuploid cells is problematic for several reasons. Most importantly, there is a substantial fraction of arrested cells present in this population that could be fully responsible for the observed clearance and no compelling evidence is presented to prove or disprove this. If indeed the arrested aneuploid cells are responsible for the cell killing in the 'cycling population', the findings are not very novel and I would therefore not support publication in EMBO J. However, if the authors can convincingly show that indeed the cycling cells are direct target cells for NK cells through NFkB, which I realize are challenging experiments, I would be supportive of publication. Please find my specific points below.

Major comments

1. The title suggests that a common mechanism for aneuploid cells is found and described. However, throughout the manuscript two very distinct groups of aneuploid cells are studied; the ArCK cells and the aneuploid cycling cells. I have some issues with the distinction/generation of these different groups.

- First, the second group is referred to as 'cycling aneuploid cells'. However, there is no data presented that specifically the aneuploid cells within this group are indeed still cycling. Importantly, of this group, 40% of the cells contain an aberrant karyotype (of an n=11 experiment) and thus 60% would have a normal karyotype. At the same time only 60% of these cells are KI-67 positive and even approximately 30 % of these cells are β -gal positive (Figure S1 and Figure 3D). This indicates that in fact a substantial fraction of this population is arrested in G1, likely with signs of senescence and this population could be completely responsible for the observed phenotypes/clearance by NK cells. Thus, there is no data provided to demonstrate that 1) there are still aneuploid cells cycling in this population and 2) that aneuploid cells that are still cycling are indeed a target for NK cell mediated clearance or if it are solely/mainly the arrested cells that are targeted. Hence, the name 'aneuploid cycling cells' is also incorrect as a large (if not all) aneuploid cells might be arrested.

- An additional problem here is that the excreted factors are shown to be sufficient to kill control cells (Figure 4). So even if cycling aneuploid cells are present and can be shown to be killed, this could be an indirect effect driven by the substantial number of arrested cells present in this population. Direct evidence would require to isolate the cycling cells only (by live FACS sorting approaches possibly like S-phase of FUCCI cells), show that this population still contains a large fraction of aneuploid cells and are efficiently killed by NK cells.

- The authors further show that the cycling cells differ from the ArCK cells in their senescent phenotype. However, importantly the ArCK cells are analyzed 156h post missegregation while the cycling aneuploid cells are analyzed 60h post missegregation. Thus, these differences might reflect the difference between early arrested cells and long term arrested cells.

All of this is not properly discussed in the paper but make the findings and their interpretation

regarding the cycling aneuploid cells rather weak. Also, the term 'cycling aneuploid cells' is therefore not suitable unless stronger evidence is shown that the cycling cells are aneuploid and are sufficient to trigger NK cell killing. This needs both experimental and textual adjustments.

We thank Reviewer #1 for her/his insightful and constructive suggestions. We appreciate that she/he acknowledged that “the finding that the NF- κ B pathway plays a central role in clearance of aneuploid cells by the immune system is interesting and would be a relevant finding to the aneuploidy field”. At the same time, she/he had major concerns on whether and how the “aneuploid cycling cells” are also cleared by NK cells. As noted above, we have removed the data referring to aneuploid cycling cells and present in the manuscript a major discovery, namely NF- κ B upregulation triggers NK cell-mediated immune clearance of aneuploid cells, which is, as pointed by the reviewer, a relevant finding.

2. A second major comment is that the main claim of the paper is that NF κ B plays a critical role in the killing of aneuploid cells. However, the experiments to proof this are incomplete. Figure 6D: The double knockouts for RELA and RELB are not used to study the effect on aneuploid cycling cells. It is unclear why they didn't do this experiment. This is a critical experiment to fully support their hypotheses. Instead only the pan-inhibitor of NF- κ B is used in the cycling aneuploid cells, however inhibitors can have side effects and nothing is done to show that the inhibitor at the used dose is indeed effective and specific for NF- κ B signaling.

We thank the reviewer for pointing out this issue. Since we have agreed with the editor to remove the data referring to aneuploid cycling cells, we did not include the *RELA RELB* double KO in aneuploid cycling cells in the revised manuscript.

We have shown that the condition we used for the NF- κ B inhibitor BMS-345541 treatment significantly reduced TNF α induced RelA and RelB nuclear translocation in RPE1-hTERT cells. The data supporting this are presented in figure EV3E. We recognize that inhibitors can have side effects. However, this finding is backed-up by our demonstration that in ArCK cells reduction in NK cell-mediated killing was also observed in two independent clones of *RELA RELB* double KO cell lines (figure 5E-F and figure EV3D). The consistency across experiments performed using both genetic knockouts and the chemical inhibitor support our conclusion that NF- κ B activation in ArCK cells is suppressed by the inhibitor treatment.

Minor comments

- Figure 1A: Quantifications of the killing of cells is solely based on manual analysis (engagement by NK cells and lifting of the cell from the plate). A more unbiased/semi-automated approach would be more convincing. Why do the authors not base their quantification on TOPRO3 entry as they use in their example images? Or another marker for cell death when filming the cells.

We thank the reviewer for pointing this out. In our co-culture assay setup, we found that the TOPRO3 background signal increases significantly as the time course proceeds, likely due to increasing amount of cell debris. This background signal often co-localized with intact target

cells over multiple time frames, which made the analysis solely by TOPRO3 signal subject to inaccuracies, especially when scoring a large number of target cells.

To prove that the analysis of NK cell-mediated killing using our approach of phase contrast imaging is consistent with measuring TOPRO3 uptake, we have now conducted a side-by-side analysis of the same movie with both quantification methods. To accurately quantify TOPRO3 signal, we used phase contrast image as a reference to distinguish the target cells from either NK cells or cell debris, and then independently measured the time when TOPRO3 was taken up by the target cells. This analysis showed that the time of NK cell-mediated killing assessed by our phase contrast imaging method agreed with TOPRO3 uptake data. These data are now presented in fig EV1A.

- The procedure to generate ArCK cells requires multiple round of nocodazole shake off. It is not unlikely that the addition of nocodazole and washout of nocodazole also induces cells to undergo mitotic slippage, resulting in a cell cycle arrest and become senescent. Did the authors ever perform this procedure without the MPS1 inhibitor treatment? i.e. would the procedure to select for the ArCK cells not result in a population of senescent cells independent of the MPS1 inhibitor?

We thank the reviewer for highlighting this important point. We have done the suggested experiment - which is to perform nocodazole shake-off on control cells without Mps1 inhibitor treatment - in our previous publication (Santaguida et al. 2017, figure 6I). This showed that euploid control RPE1-hTERT cells without Mps1i treatment were completely removed from the plate after two consecutive nocodazole shake-offs, indicating that nocodazole treatment along did not lead to senescence in our experimental setting. The corresponding data are also shown below as a reference [Figure for referees not shown.] .

In addition, we have added the following sentence in the manuscript to clarify this issue: “Importantly, such cell cycle arrest was not due to prolonged nocodazole treatment since euploid control cells were completely removed after two consecutive rounds of shake-offs (Santaguida et al., 2017)”. Page 4, third paragraph.

- Figure 2: The authors claim that both G1 arrest as well as signs of senescence contribute to cell killing in aneuploid cells. This is based on the comparison to other treatments inducing a NK

response. I have some issues with these experiments. First of all, comparing different treatments to each other works to exclude certain traits as immune response inducers but it is not sufficient to argue about the minimal requirements in the aneuploid setting. Furthermore, all conditions where NK cell killing is induced are conditions where cells were treated for a prolonged amount of time (doxorubicin, palbociclib and nutlin3, 7 day treatment). On the contrary, the authors claim that G1 arrest without signs of senescence is not sufficient. For this, the authors switch to Torin. However, this treatment is only done for 24h which is likely not sufficient to induce a senescent phenotype. This raises the question why the authors did not choose to treat their cells with the original inhibitors (for example palbociclib) which would also induce a G1 arrest upon 24 hours but is likely not sufficient to drive a senescent phenotype. Since the inhibitors and way of inducing the arrest would be comparable between palbociclib 24h and palbociclib 7 days it is more valid to discriminate the contribution of the senescence (and exclude that a G1 arrest is sufficient). Vice versa could also be done, what is the effect of 7 days of torin treatment? Do the cells become more senescent and start to induce NK cell killing?

We thank the reviewer for these suggestions aimed at improving consistency across all the drug treatments. Based on her/his suggestion, we have replaced all the data regarding our original condition with torin1 treatment (1 day, 5 μ M) with a 7 day treatment (1 μ M, we decreased drug concentration since prolonged torin1 treatment using the original drug concentration led to high cell toxicity).

We are now showing that the majority of cells were arrested in the cell cycle after 7 days of torin1 treatment as judged by EdU incorporation assay (see figure 2F). When we performed NK cell-mediated killing on these cells, we still found that NK cell-mediated killing in torin1 treated cells was comparable to that of control cells (figure 2G, KS test, $p = 0.79$, not significant). We found after 7 days of torin1 treatment, cells arrest in quiescent state instead of senescence, as judged by low beta-gal positive staining (now figure 3C and EV2B). We conclude that G1 arrest in target cells contributes to NK cell engagement but only when accompanied by features of senescence (please see page 6, third paragraph).

- Page 4, last paragraph. Unclear description: the ratios of NK cells and target cells are changed both for euploid cells as well as ArCK cells; whereas from the text it seems like only the ratio between NK cells and ArCK cells is altered. From the text it should be clear that for both conditions, the NK cell ratio is increased. This part should be rewritten for better understanding.

We apologize for the lack of clarity. In all panels, the ratio between the NK cell and target cell is always the same for both control and ArCK cells. We have clarified this in the text by changing the text to “We found that at a ratio of 2.5 NK cells to 1 target cell, ArCK cells were consistently killed twice as effectively as euploid control cells.” (page 4, last paragraph).

- Figure 2: Nutlin treated cells partially start to proliferate after drug release as seen by EdU incorporation and increased cell number; but are most effectively killed. It indeed has been previously reported that Nutlin3 induces a senescence-like phenotype that is reversible (as reported here) but in fact it was shown to reduce SASP signaling as cells display reduced NF κ B signaling (<https://doi.org/10.1038/s41598-018-20000-4>). The highly efficient killing by NK cells

is thus very unexpected. Is this cell killing also rescued by the inhibition of NFκB? Or is another pathway involved in the case of Nutlin?

The reviewer pointed out that our finding on nutlin3 treated cells (which induced a reversible senescent-like phenotype) is consistent with previous reports. In addition, based on our RNA-seq data, NF-κB pathway is not upregulated in nutlin3 treated cells. To investigate whether the unexpected high level of NK cell-mediated killing will also be abolished by inhibiting NF-κB pathway. To test this, we performed NK cell-mediated killing by treating *RELA RELB* KO cells with nutlin3. We found that inactivating NF-κB pathway in nutlin3-treated cells did not significantly affect NK cell killing. This suggest that other pathways - beside NF-κB - might be involved in triggering killing of nutlin3-treated cells. We did not include these data into the manuscript because we prefer to make aneuploidy the major focus of the paper. We have included the data below for reviewer's reference [Figure for referees not shown.] .

- Page 7, paragraph 3: the statement 'aneuploid cycling cells exhibited only slightly increased levels of SA-beta-gal positive cells compared to the untreated proliferating population' should be adjusted as 30% of cells are SA-beta-gal positive, which is not a slight increase considering that only 40 % of the cells show an aneuploid karyotype (see major comment above).

As mentioned above, we have removed the data regarding aneuploid cycling cells.

- Figure 3E: No Torin is added for IFN-a and IFN-b. Is there a reason for that?

We have done the ELISA assay assessing IFN-a and IFN-b levels in cells treated with torin1 (1 μM) for 7 days. We did not see high degrees of IFN-a or IFN-b secretion in the torin1 treated cells. The data are now included in figure 3D.

- Figure S4: Large variations in the SEAP assay are found. Generating a stable cell line expressing

this reporter would possibly help to evaluate the level of NF- κ B activation compared to transient transfection. This would also allow to perform this assay for both the ArCK cells the cycling aneuploid cells.

We agree that there is large variations and appreciate the reviewer's suggestion. On considering this issue, we realized that the main obstacle to the measurement of SEAP in ArCK cells is related to the procedure utilized to generate these cells, which employs multiple rounds of shake-off and medium removal, and thus leads to loss of medium in which SEAP is present. Thus, to circumvent this problem, we have repeated the experiment but now measuring AP secretion 96 hours post reversine-induced chromosome mis-segregation. This is a timepoint reachable without cell splitting and, at the same time, reasonably close to the time required for ArCK collection, thus allowing us to collect the AP conditioned medium without cell splitting or repetitive medium changing due to the shake-off process. With this modified experimental protocol, we observed a three-fold increase in SEAP level in aneuploid cells compared to their euploid counterparts, with much more consistent agreement between replicates, which indicated NF- κ B to be activated in ArCK cells. These data are now presented in figure 4E and described in page 8, second paragraph.

- Figure 5D. On several occasions in the paper it is not indicated what the error bars represent. In this figure the raw data is present in the supplemental data and based on those numbers the error bars do not seem to represent standard deviations which would be appropriate here. Please indicate everywhere what the error bars represent.

Thanks for the suggestion. We are now indicating in the figure legends what the error bars represent. The SEAP data is now presented in figure 4E (not 5D) and in the corresponding figure legend we have indicated: "n=3; mean \pm SEM. unpaired t-test, p<0.0001".

- Figure 6A: The light grey line is the same in both plots: A. Cyc. Control and ArCK control. A. Cyc. are not killed so effectively so the line is likely to be wrong in the left plot.

The grey lines in A. Cyc. Control and ArCK control were the same, because the experiments assessing A.Cyc cell killing and ArCK cell killing were performed side by side. However, since we are removing the data regarding aneuploid cycling cells, we have removed the killing results in the left panel. The result on ArCK cells is now presented in figure 5A.

- In some plots (Figure 6A left and right plot, Figure S5C left and right plot, Figure 6B and Figure S5D) the controls lines are identical. This is not necessarily a problem in case these experiments have all been done simultaneously. This is important as the extent of cell killing is quite variable and thus internal controls are always required. Are these lines indeed from the same experiments? The authors should indicate that somewhere.

Yes, the control lines are identical because the two experiments were conducted simultaneously as the reviewer speculated. We are indicating this in the corresponding legends

for figure EV3D: "The same controls were plotted in (D) and figure 5E since all three replicates for both *RELA RELB* KO clones were performed side by side".

- Figure 6D and Figure S5D. Why are there no error bars? Is this an n=1? This needs repetition.

We have repeated the experiments with at least two biological replicates for each condition. The data are now presented in fig 5G and EV3C. As per journal guidelines, we are including scatter plots.

- Figure 6D. The control cells are also more effectively killed in the ArCK treated with the pan-inhibitor of NF- κ B; up to 80% is killed by NK cells, where in other experiments this was between 30-60%. If the control of the A. Cyc experiment would be plotted, there would even be additional cell killing by the addition of the NF κ B inhibitor.

Because the NK cell-mediated killing assay is based on analysis from single cell measurements, we recognize that there are more variations between experimental replicates. We have always included euploid control cells in the killing assays to assess the baseline of killing of RPE cells. We have included two more experimental replicates for the NK cell-mediated killing results presented in figure 6D (Now figure 5G). We still see a significant reduction in ArCK cells when treating with the NF- κ B inhibitor BMS-345541. We have removed the data referring to aneuploid cycling cells.

- Figure S7. Although the authors state in the text that deletion of *STAT1* does not significantly affect the NK cell killing, some experiments are in fact statistically significant (clone 2, both cycling and ArCK cells).

We agree that variations in statistical significance exist between individual *STAT1* KO clones. We think such variation could be due to the fact that NK cell killing assay was analyzed at the single cell level. To confirm the conclusion that we made on *STAT1* KO ArCK cells, we have repeated the NK cell killing assay on all three *STAT1* KO clones. All four experimental replicates are now included in the graph. The difference between NK cell-mediated killing in ArCK cells generated in control and *STAT1* KO background in clone 1 and 3 indeed did not reach statistical significance level (now figure EV4E, KS test, ArCK-c1 vs. ArCK *STAT1*-c1, $p=0.7$, *n.s.*; ArCK-c1 vs. ArCK *STAT1*-c3, $p=0.14$, *n.s.*). The significance in clone 2 is right on the boarder of the significance (ArCK-c1 vs. ArCK *STAT1*-c2, $p=0.05$). For the KS test we employed, significant differences between conditions often leads to a p-value much lower than 0.05. We think the results are consistent between different clones.

- Page 11 and figure S8: Why are these experiments only done in the discussion? These are relevant experiment so I would suggest to move them to the result section and to be discussed in the discussion section.

We thank the reviewer for these suggestions. We have moved the text regarding these data

from the discussion to the results under the section “Multiple signals trigger immune clearance of aneuploid cells” (please see page9 paragraphs 2 and 3). The data are now presented in Figure EV5.

- Figure S5D: typo in last graph on the right: ArCK RELB-c3 instead of c32
We have fixed this.

- Page 6: typo: Figure 2a)
We have fixed this.

Referee #2:

The manuscript by Wang RW et al. provides novel mechanistic insight into the process of NK cell-mediated immune clearance of aneuploid cells following up their previous findings. A series of elegant experiments demonstrate that the NF- κ B pathway is required for the NK cell-mediated killing of aneuploid cells. Cell cycle arrest is not a prerequisite for aneuploid cell recognition by NK cells, although senescence triggering potentiates the recognition. Both cell autonomous and non-cell autonomous signals are involved in this immune clearance, even though the molecular characterization of the NF- κ B activating signals will demand deeper future analysis. Moreover, the authors found that highly aneuploid cancer cell lines still retain the NF- κ B signaling response.

The manuscript is of broad interest regarding the understanding of cancer cell immune evasion, and of significant impact to the aneuploidy field. I am very supportive of publication in the EMBO Journal provided that the authors address specific comments in order to strengthen their conclusions.

We thank the reviewer for her/his comments and for recognizing the importance and novelty of our work.

Major comments:

1. The authors start by showing that aneuploid senescent cells (ArCK) are killed two times more effectively than euploid controls. Considering that RPE-hTERT cells express hTERT and have chromosome 10q aneuploidy, the authors also tested another cellular model (fibroblast primary cultures). Interestingly, they found large variations in NK cell-mediated killing. Nevertheless, to further extend their findings using RPE1 the authors could test one of the following options.

1.1. Would aneuploidy increase HDF-Ad recognition by NK cells as in the epithelial cell model?

As suggested, we have now assessed NK cell-mediated killing in the Mps1 inhibitor reversine induced aneuploid cells generated in NHDF-Ad cells. Similar to what we have observed in RPE1-hTERT cells, we saw a significant ~2 fold increase in NK cell-mediated killing in aneuploid NHDF-Ad cells compared to their euploid control. We have added these data to the manuscript as figure EV1E (NHDF-Ad Ctrl vs. NHDF-Ad Mps1 inhibitor, $p= 0.001$, KS test).

1.2. Alternatively, the authors could test whether the HDF-Neo immunogenicity is dependent on NF- κ B activation (e.g. treatment with pan-NF- κ B inhibitor).

As suggested, we have now assessed NK cell-mediated killing in NHDF-Neo cells with or without the NF- κ B inhibitor BMS-345541. We did not see a significant decrease in the NK cell-mediated killing in inhibitor treated NHDF-Neo cells compared to their euploid control cells. These data are also shown in figure EV3F (NHDF-Neo Ctrl vs NHDF-Neo NF- κ B inhibitor, $p= 0.28$, *n.s.*, KS test), indicating that other signals, beyond NF- κ B activation, are responsible for basal immunogenicity of those cells.

2. One interesting finding is that cell cycle arrest per se (unless accompanied by features of senescence) is not enough for NK cell recognition. Inactivation of the TORC1 pathway was a key elegant experiment to support this conclusion. Regarding this assay a few comments.

2.1. In comparison to other drug treatments used to induce G1 arrest (doxo, palbo and nutlin3) for 7 days, the mTOR kinase inhibitor (torin1) treatment was used for 1 day. Could the authors use similar conditions (most likely 1 day)?

We appreciate the reviewer for experimental suggestions on the consistency of all the drug treatments. Instead of treating cells with doxorubicin, palbociclib, or nutlin3 for 1 day, we replaced our original condition for 1 day torin1 treatment with 5 μ M with 7-day treatment with 1 μ M. Note we had to decrease the drug concentration because we found a significant increase in cell death with prolonged torin1 treatment using the higher drug concentration for a prolonged time. This lower concentration still gives robust G1 arrest but does not trigger senescence, as judged by the absence of beta-gal positivity (Fig 3C and EV2B). In this way, the timing of all drug treatment is now comparable.

2.2. The authors should include torin1 treatment condition in the gene expression analysis shown in Supplementary Fig. 3 to demonstrate that only G1 arrested cells with senescence features are able to elicit NK cell-mediated cytotoxicity.

As requested, we have performed RNA sequencing and gene expression analysis on cells treated with torin1 for 7 days. The data are now included in figure 4A and EV2D. Interestingly, we do see the NF- κ B signature in torin1 treated cells, but NF- κ B upregulation did not lead to a significant increase in NK cell-mediated killing in torin1 treated cells (figure 2G, KS test, $p = 0.79$, not significant). Since torin1 treatment leads to quiescence but not senescence, this result agrees with our conclusion that G1 arrested cells associated with features of senescence can elicit NK cell-mediated cytotoxicity.

3. In page 6, 'Aneuploidy causes NK cell recognition by mechanisms in addition to triggering senescence'. This subtitle is somehow misleading in the sense that even A.Cyc cells exhibit increased senescence markers. Senescence is an evolving phenotype that only ultimately leads to irreversible cell cycle arrest. All the conditions tested end up having increased senescence markers: DNA damage and SA-b-gal levels correlating well as expected (doxo>ArCK>ACyc); and the SASP secretome heterogenous as expected for the different stresses, although almost detectable in A.Cyc cells. Since A.Cyc pre-conditioned medium induces NK cell-mediated cytotoxicity in euploid cells, I suggest the authors to explore the gene expression analysis to show that the core senescence-associated signature (Hernandez-Segura et al. Curr Biol 2017) is increased in the A.Cyc cells circumventing the limited sensitivity of the cytokine array. This would better guide the readers to the following section and Fig. 4.

We appreciate the reviewer for pointing out this issue and agree with his/her nuanced reviews . We recognize that there is a fraction of arrested cells presented in the aneuploid cycling cells and it will be challenging to separate out a pure proliferating aneuploid population using our experimental scheme. Consequently, with agreement from the editor, we have decided to remove all the data referring to aneuploid cycling cells and have the paper considered for EMBO Reports not EMBO Journal.

4. As appropriately concluded in page 8, " NK-cell mediated killing of aneuploid cells is largely mediated by secreted factors, but cell surface also contribute, especially in the case of ArCK cells". To support this conclusion, the authors should provide a comparative analysis of cell surface markers between ArCK and A.Cyc.

We thank the reviewer for suggesting these important experiments. We agree that profiling and analyzing the cell surface markers in ArCK and A.Cyc is definitely an interesting field of study. However, since we have decided to remove all the data referring to aneuploid cycling cells and given the complexity of those experiments, we have decided not to perform those studies for this manuscript. We have also changed the text to make a more accurate conclusion based on our current data: "We conclude that conditioned medium provides one or more factors that upregulate NK cell-mediated killing and that aneuploid cells could generate both cell autonomous and non-cell autonomous signals that render them susceptible to NK cell-mediated cytotoxicity" (now page 7, second paragraph).

5. In my opinion, the data shown in Supplementary Fig. 8 should be described during the results section. For instance, Sup. Fig. 8a,b fits well in the section on the canonical and non-canonical NF- κ B pathway (RelA/RelB KO, STAT1 KO). Sup. Fig. 8c-e brings interesting novel data on retrotransposon signalling for NK- κ B activation, which perhaps together with a more throughout analysis of surface molecules could be a stronger Fig. 7 to finish the manuscript with rather than the still very preliminary analysis of aneuploid cancer cell lines.

We thank the reviewer for the suggestions. We have moved the text regarding these data from

the discussion to the results under the section “Multiple signals trigger immune clearance of aneuploid cells” (please see page9 paragraph 2 and 3). We kept the figures associated with these data in supplement (now figEV5) because we would like to make the connection of NF- κ B signaling in cancer cell lines in the main figure (figure 6).

Minor comments:

1. In Fig. 1 the authors should reorganize the graphs layout so that graphs b and c are side by side, then graphs d below, and finally graphs e and f side by side.

We have fixed this.

2. Data on Fig. 3e (secreted cytokine levels GM-CSF, MIF, CCL2) are also included in Supplementary Fig. 2c.

We have removed the duplicated data in figure EV2C.

3. Fig. 4b should alternatively be Fig. 4a. Also, the authors should mention somewhere (in the legend perhaps) that the Ctrl data are the same in both graphs. This also applies to Fig. 6a (Ctrl-c1 and CtrlRELA-c1), Fig. 6c (Ctrl-c1 and ArCK-c2) and to Supplementary Fig. 5c,d.

Yes, the controls are the same because these experiments were performed simultaneously. As explained previously, we have decided to remove the data regarding aneuploid cycling cells. For figure 6c (now figure EV3D), we have indicated in the figure legends “The same controls were plotted in (D) and figure 5E since all three replicates for both *RELA RELB* KO clones were performed side by side.”

4. In Fig. 5b an enrichment plot for "Epithelial-mesenchymal-transition" signature is shown but poor description or rationale is given in the text (page 8) for this particular analysis.

We agree. We have removed this and replaced it with interferon signatures in ArCK cells, which are more relevant for this study.

5. In Fig. 6d, standard deviations are missing in the graph on the right (ArCK- NF κ B inh experiment). The same happens in Supplementary Fig. 5d (ArCK-RELB c3 experiment).

We have now included more experimental replicates. For figure 6D (now fig 5G), individual data points as well as mean and SEM are presented. For figure S5D (now fig EV3C), individual data points and mean are presented on the graph as per journal guidelines.

Referee #3:

Chromosome segregation is tightly regulated during cell division to prevent aneuploidy, which is detrimental for normal cell viability. However, and in contrast to normal cells, cancer cells often display high levels of aneuploidy. Aneuploidy has been shown to promote tumorigenesis and thus mechanisms to eliminate and/or prevent the proliferation of aneuploid cells exist. Previous work from Santaguida and colleagues showed that normal cells with complex karyotypes (ArCK cells, highly aneuploid cells) are eliminated by NK cells in vitro. However, the mechanisms that lead to this elimination were unclear.

Here the authors take advantage of the previous published system to induce ArCK cells in the normal cell line RPE-1 to dissect the mechanisms by which NK mediated elimination occurs. They found that indeed ArCK cells are more targeted by the NK cells than the euploid counterparts and thus increase cell death is observed. In addition, cycling aneuploid RPE-1 cells seem to be also targeted by NK cells but to a lesser extent. While cell cycle arrest does not seem to be essential to induce NK killing, G1 arrest independently of aneuploidy can also promote NK killing in conditions where this arrest leads to senescent-like phenotype. Thus, senescent-like arrest in ArCK cells could be the main reason for their targeting. Conditioned medium collected from ArCK cells is sufficient to increase NK killing of euploid cells, suggesting a non-cell autonomous role in this process. While the factors have not been identified, the authors showed that activation of NF- κ B in ArCK cells is important for NK killing.

Overall this is an interesting manuscript that follows up from the authors previous work. Experiments are generally well controlled and data of high quality. Although I believe this work will be of interest to a broad readership, the significant advance made here is not as clear. Furthermore, there are several concerns that should be addressed prior to publication, which I outlined below:

We thank the reviewer for finding our manuscript interesting, with well controlled experiments and data of high quality. She/he is also outlining important points to be addressed and we thank her/him for providing important suggestions that we are addressing below.

#1. Similar to the authors previous work, here they use the RPE-1 cell line as model system. While this is a valid system to look at normal cell physiology, it would be important to assess the robustness of this phenotype in other non-cancer cell lines. For example, in supplementary figure 1d the authors demonstrate that the primary cells NHDF-Ad are somewhat resistant to NK killing and thus these cells could be used to test if complex karyotypes could promote NK killing. This is particularly important considering that NK killing varies with different cell lines (supplementary figure 1d).

We thank the reviewer for this suggestion. We have assessed NK cell-mediated killing in the Mps1 inhibitor reversine-induced aneuploid cells generated in NHDF-Ad cells. Similar to what we have observed in RPE1-hTERT cells, we found a significant ~2 fold increase in NK cell-mediated killing in aneuploid NHDF-Ad cells compared to their euploid control. We have added this data in the revised manuscript as fig EV1E (KS test, NHDF-Ad Ctrl vs. NHDF- Ad Mps1

inhibitor, $p= 0.001$), and included following sentence in the text “Importantly, we also observed a consistent two-fold increase in killing on the Mps1 inhibitor reversine-induced aneuploid NHDF-Ad cells compared to their euploid control cells (fig EV1E), indicating such NK cell-mediated immune clearance on aneuploid cells is not a cell type specific phenotype” (page 5, second paragraph).

#2. The authors propose that aneuploid cycling cells can also be targeted by NK cells, thus they conclude that cell cycle arrest is not the only requirement for this. However, I do think the data presented here demonstrates that. In fact, the data presented in figure 2h clearly shows that this aneuploid cycling cells have a decrease in cell proliferation as assessed by Ki67 staining (~40% reduction). Thus, the authors cannot exclude the possibility that the arrested cells within that population (Ki67 negative) are the ones being targeted by the NK cells. The authors could quantify the % of Ki67 positive and negative cells that are targeted by the NK cells. Alternatively, they could look into either proliferating aneuploid cancer cells or RPE-1 cells p53^{-/-} and test if cells that do not arrest in response to aneuploidy are still recognised by NK cells and killed. Without that they cannot conclude that cycling cells are being targeted.

We thank the reviewer for pointing out the potential issue of the aneuploid cycling cells. We recognize that there is a fraction of arrested cells present in the aneuploid cycling sample and it will be challenging to separate out a pure aneuploid cycling population based on our current experimental scheme. We appreciate the reviewer for suggestions on potential experiments to address this question. Although we think this is an important issue, unfortunately we are not in a position to address this point properly. With agreement from the editor, we decided to remove the data referring to aneuploid cycling cells and to focus on strengthening the finding related to NF- κ B activation in ArCK cells and its role in triggering NK cell-mediated immune clearance.

#3. Related to the point above, although the ArCK cells have been characterized in their previous paper, it would be helpful to compare side by side the karyotypes of ArCK and cycling aneuploid cells in this manuscript, as it is likely that there is a lot variability that needs to be accounted for with these treatments. Here the authors show that 45% of cycling aneuploid cells have copy number alterations but in their previous paper the ArCK cells seem to have much less copy number alterations (~10%). Thus, considering this variability, it would be great to compare these two populations here.

As explained above, we have removed the data on cycling aneuploid cells.

#4. Related to the above comments, do the authors know if certain types of aneuploidies are more targeted by NK cells? In a way that this could help selecting certain karyotypes during tumour evolution?

This is definitely a very insightful experiment but, in our opinion, is beyond the scope of this study.

#5. It is unclear if NK cells specifically targeted aneuploid cells or senescent-like cells. Firstly, the authors showed that other insults leading to arrested senescent-like can also lead to NK killing. Secondly, in the case of cycling aneuploid cells, the authors cannot exclude that few senescent like cells with the population could play a role in the non-cell autonomous activation of NK cells. In supplementary figure 2b some senescent cells (b-galactosidase positive) can be observed in cycling aneuploid cells. Thus, without additional evidence I do not think the data presented here can exclude that SASP components play a role in aneuploid mediated cell death and perhaps NK cells cannot recognise aneuploid but rather senescent-like cells.

This is a very interesting point, we agree that there is a fraction of arrested/senescent aneuploid cells present in the aneuploid cycling sample. As described previously, it is challenging to isolate a pure aneuploid cycling population using our system. In fact, we think in untransformed cells, the induction of aneuploidy is often associated with features of senescence (*e.g.*, DNA damage, proteotoxic stress, increased cytokine secretion, etc - please see discussion on this issue in the manuscript, page 11). We think these features indeed contribute to NK cell-mediated killing.

Therefore, in untransformed cells, it is difficult to distinguish the contribution of senescent from aneuploidy *per se* on NK cell-mediated killing.

#6. While different insults that lead to arrested senescent-like cells promotes NK killing, only aneuploid and Doxo treated RPE-1 cells have increased NK-kB gene expression signatures. Is it possible that DNA damage downstream of aneuploidy and Doxo treatment is the cause for NF-kB killing? And could the authors test of RelA/B KO Doxo treated cells escape NK killing? Additionally, do arrested cells that are killed by NK cells, but do not show increased NF-kB signatures (*e.g.* nutlin treated cells) also require RelA/B? This would demonstrate the specificity of this the NF-kB pathway to aneuploid cell killing.

We thank the reviewer for the experimental suggestions. We have performed NK cell killing assay in doxorubicin or nutlin3 treated cells in *RELA RELB* KO. We found that inactivating NF-kB pathway in doxorubicin treated *RELA RELB* KO cells led to a modest but significant decrease in NK cell-mediated killing. On the other hand, *RELA RELB* KO did not rescue NK cell-mediated killing of nutlin3 treated cells.

These results suggest, as the reviewer speculated, that DNA damage downstream of aneuploidy could be involved, at least in part, in eliciting NK cell-mediated killing in ArCK cells. On the other hand, pathways other than NF-kB might be involved in the NK cell-mediated killing of nutlin3 treated cells. We did not include these data into the manuscript because we want to focus mainly on aneuploid cells in our study, but we have included the data below for reviewer's reference [Figure for referees not shown.] :

#7. The authors mention that the conditioned medium contain factor that attract NK cells. However, it is more likely that the secreted factor activates NK cells since in the conditioned medium experiments any gradient that could attract cell would be lost. I suggest re-phrasing this.

Since we are using NK92-MI cells which has been activated by constitutive IL2 expression, we think it is more likely to be additional factors presented in the conditioned medium that upregulate the activation signal for a higher NK cell-mediated killing. However, we do not have data to address this. Thus, we have rephrased the text to be more nuanced as following: "We conclude that conditioned medium provides one or more factors that upregulate NK cell-mediated killing and that aneuploid cells could generate both cell autonomous and non-cell autonomous signals that render them susceptible to NK cell-mediated cytotoxicity." (page 7, second paragraph).

#8. It would be helpful to look at AP secretion or SEAP reporter assay in both cycling and

arrested aneuploid cells. In principle the expectation is the NF- κ B is more activated in arrested ArCK cells, but this was not shown.

On considering this issue, we realized that the main obstacle to the measurement of SEAP in ArCK cells is related to the procedure utilized to generate these cells, which employs multiple rounds of shake-off and medium removal, and thus leads to loss of medium in which SEAP is present. Thus, to circumvent this problem, we have repeated the experiment but now measuring AP secretion 96 hours post reversine-induced chromosome mis-segregation. This is a timepoint reachable without cell splitting and, at the same time, reasonably close to the time required for ArCK collection, thus allowing us to collect the AP conditioned medium without cell splitting or repetitive medium changing due to the shake-off process. With this modified experimental protocol, we observed a three-fold increase in SEAP level in aneuploid cells compared to their euploid counterparts, with much more consistent agreement between replicates, which indicated NF- κ B to be activated in ArCK cells. These data are now presented in figure 4E and described in page 8, second paragraph.

#9. Despite the NF- κ B gene expression signatures associated with aneuploid cells, the authors could not see any evidence of RelA/B accumulation in the nucleus. This was also the same for the LPS positive control. Could the authors comment on this? Is the assay not working? Or it does not work in RPE cells? Perhaps testing in a different cell line?

We agree with the reviewer that nuclear accumulation of RelA/B should be present at least in LPS-treated samples. To address this, we took advantage of an assay for detection of nuclear RelA/B that was recently employed in cells treated with Mps1 inhibitors (Vasudevan et al., 2020), namely assessment of RelA nuclear translocation by immunofluorescence staining. By doing so, we observed a significant increase in the nuclear RelA signal in ArCK cells compared to the control (figure EV3A). Moreover, we have now validated the upregulation of RelA and RelB target genes by RT-qPCR (figure 4D). In addition, the SEAP reporter assay provides evidence for NF- κ B activation in ArCK cells (figure 4E).

#10. I could not find the data of RelA/B double KD in cycling aneuploid cell killing? Was that tested?

As noted above, we have removed the data referring to aneuploid cycling cells from the manuscript.

#11. The link to cancer is not clear. How the degree of aneuploidy was assessed in the CCLE cell lines should be described, it makes it difficult to interpret the data in figure 7. Furthermore, could the authors show a correlation plot to see if indeed there is a correlation between levels of aneuploidy and NF- κ B in cancer cell lines? This will make the point clearer.

The method for assessing the degree of aneuploidy is published in (Cohen-Sharir et al., 2021). We now referred to the method by citing the paper on page 10, first paragraph "The degree of aneuploidy was scored in almost 1000 cell lines in the CCLE as described in (Cohen-Sharir et al.,

2021)". We have also included the requested correlation plot in figure 6B – this correlation is indeed significant (Spearman's $\rho = 0.181$, $p\text{-value} = 2.2e-08$).

#12. Related to the point above, could the authors test if induction of aneuploidy in cancer cells also lead to NF- κ B activation and NK killing? For example, using the stable cell line HCT-116? Are cancer cells less sensitive because they do not arrest in response to aneuploidy?

We thank the reviewer for raising this interesting question about the cancer cell lines. We induced aneuploidy in HCT-116 and DLD-1 cancer cell lines (both are pseudo-diploid) by reversine treatment. In agreement with previous reports, we found that in HCT116 and DLD-1 cells, Mps1 inhibitor treatment induced NF- κ B pathway upregulation (Vasudevan et al, 2020). The data are presented in figure EV5F. Despite this, we did not observe an increase in NK cell-mediated killing upon Mps1 inhibitor treatment in both cancer cell lines. We speculate that NK cell-mediated immune clearance promoted by aneuploidy through NF- κ B may only occur in untransformed cells or during early stage of tumorigenesis. Whereas in cancer cell lines, which often have undergone clonal evolution, NK cell-mediated immune response is perhaps weakened. This is probably due to the fact that cancer cells behave differently from untransformed cells in response to aneuploidy induction and as the reviewer suggested, cell cycle arrest does not take place. We have also included this speculation in the discussion section in the revised manuscript on page 11, first paragraph.

#13. Results of the supplementary figure 8 should be described in the results section. It is very difficult to conclude that up-regulation of transposons could play a role in NK killing since the differences upon treatment with 3TC do not seem to be significant.

We thank the reviewer for this suggestion. We have moved the text regarding these data from the discussion to the results under the section "Multiple signals trigger immune clearance of aneuploid cells" (page9, paragraphs 2 and 3). We agree that the difference upon treatment with 3TC in ArCK cells is mild, however, at the same time, it's statistically significant (figure EV5E. KS test, ArCK vs. ArCK-3TC, $p < 0.0001$).

#14. The authors should not write the legends for all proliferation graphs vertically, it is very difficult to read.

We have fixed this.

#15. Missing references in the discussion: "Multiple studies have shown that senescent cells are recognized and eliminated by NK cells *in vitro*".

We have added the references here, please see page 10, third paragraph "Multiple studies have shown that senescent cells are recognized and eliminated by NK cells *in vitro* (Iannello et al., 2013; Sagiv et al., 2016; Soriani et al., 2009, 2014)". If the reviewer had other references in mind, we would be pleased to hear them.

Dear Stefano,

Thank you for submitting your revised manuscript. It has now been seen by two of the original referees. As you can see, the referees find that the study is significantly improved during revision and recommend publication. Before I can accept the manuscript, I need you to address the additional points below:

- Please address the remaining minor concerns of the referees.
- Please provide 3-5 keywords for your study. These will be visible in the html version of the paper and on PubMed and will help increase the discoverability of your work.
- As per our format requirements, in the reference list, citations should be listed in alphabetical order and then chronologically, with the authors' surnames and initials inverted; where there are more than 10 authors on a paper, 10 will be listed, followed by 'et al.'. Please see <https://www.embopress.org/page/journal/14693178/authorguide#referencesformat>
- We noticed that Figures EV2B, EV3B and EV3C are currently not called out in the text.

Thank you again for giving us to consider your manuscript for EMBO Reports, I look forward to your minor revision.

Kind regards,

Deniz

--

Deniz Senyilmaz Tiebe, PhD
Editor
EMBO Reports

Referee #1:

Remarks to the Authors:

The manuscript by Wang RW et al. provides novel mechanistic insight into the process of NK cell-mediated immune clearance of aneuploid senescent cells (referred to as Arrested with Complex Karyotypes, ArCK) following up their previous findings. Experimental evidence supported by state-of-the-art methodologies is provided demonstrating that both the canonical and non-canonical NF- κ B pathway are required for the NK cell-mediated killing of ArCK cells. Prolonged G1 arrest with senescence features is shown as needed for ArCK cell recognition by NK cells. Both cell autonomous and non-cell autonomous signals are involved in this immune clearance, and initial molecular characterization of NF- κ B activating signals is included. Moreover, the authors show a correlation between NF- κ B upregulation and the degree of aneuploidy in cancer cell lines, even though found insufficient to trigger NK cell-mediated clearance.

The findings meet the novelty and significance for publication in EMBO Reports. The authors have extensively addressed major concerns raised by the Reviewers during the first round of revision. It is my opinion that the revised version of the manuscript benefited from the exclusion of data referring to 'aneuploid cycling cells'. The conclusions are now well supported by the data and still

significant. The structure and flow of the manuscript have improved. I am contented that the authors accomplished to answer most of my previous comments, including novel convincing data. I am supportive of publication in EMBO Reports, although I have a few comments.

Major comment:

1. The authors should always refer to 'ArCK' or 'aneuploid senescent cells' and not simply to 'aneuploid cells' for clarity. It is the aneuploidy-induced senescence that contributes to NK-kB signalling and NK cell-mediated clearance.

This applies to the title, subtitles in pages 6, 7 and 8, subtitles in the discussion section and synopsis highlights.

Minor comments:

1. The authors should perhaps mention to the p53 pathway activation (Fig. 4A) as a common hallmark in all senescence inducing conditions (ArCK, Doxo, Palbo and Nutlin) that exhibited NK cell-mediated cytotoxicity, which is absent in the Torin quiescence condition.

2. In Fig. EV4 A, show legend 'IFN target genes' (instead of ArCK) as in Fig. 4C for 'RELA RELB target genes'.

3. In Fig. EV5, rearrange the figure distribution so that A and B are side by side (STING related panels).

4. I suggest to rephrase the subtitle in page 8 'Multiple signals trigger immune clearance of aneuploid cells'. The authors excluded the IFN alpha and gamma pathways, as well as the cGAS-STING pathway, as significantly contributing to NK-cell mediated immunogenicity of ArCK cells. Retrotransposon activation on the other hand was found as triggering signal. So, the subtitle should better highlight this finding.

Referee #2:

The authors addressed my main concerns adequately and overall I believe this work is much improved. In addition, removing the section comparing cycling and arrested aneuploid cells helps since it was still too preliminary to make strong conclusions. I have only minor comments for the authors at this stage and would support the publication of this work in EMBO Report.

Related to previous comment #6. The new experiments that the authors have included in the rebuttal letter should be included in the manuscript for 2 reasons. 1) they show that DNA damage response could be important to elicit NFkb signalling the authors observed and 2) other pathways play a role in this process. In my opinion these are important pieces of data that highlight the complexity of aneuploid cell killing.

Related to previous comment #9. The new experiments are very nice indeed. But I would encourage the authors to include images to exemplify what they quantify in EV3A.

The authors have addressed all minor editorial requests.

Dear Stefano,

Thank you for submitting your revised manuscript. I have now looked at everything and all is fine. Therefore, I am very pleased to accept your manuscript for publication in EMBO Reports.

Congratulations on a nice work!

Kind regards,

Deniz

--

Deniz Senyilmaz Tiebe, PhD
Editor
EMBO Reports

--

At the end of this email I include important information about how to proceed. Please ensure that you take the time to read the information and complete and return the necessary forms to allow us to publish your manuscript as quickly as possible.

As part of the EMBO publication's Transparent Editorial Process, EMBO reports publishes online a Review Process File to accompany accepted manuscripts. As you are aware, this File will be published in conjunction with your paper and will include the referee reports, your point-by-point response and all pertinent correspondence relating to the manuscript.

If you do NOT want this File to be published, please inform the editorial office within 2 days, if you have not done so already, otherwise the File will be published by default [contact: emboreports@embo.org]. If you do opt out, the Review Process File link will point to the following statement: "No Review Process File is available with this article, as the authors have chosen not to make the review process public in this case."

Should you be planning a Press Release on your article, please get in contact with emboreports@wiley.com as early as possible, in order to coordinate publication and release dates.

Thank you again for your contribution to EMBO reports and congratulations on a successful publication. Please consider us again in the future for your most exciting work.

THINGS TO DO NOW:

You will receive proofs by e-mail approximately 2-3 weeks after all relevant files have been sent to our Production Office; you should return your corrections within 2 days of receiving the proofs.

Please inform us if there is likely to be any difficulty in reaching you at the above address at that

time. Failure to meet our deadlines may result in a delay of publication, or publication without your corrections.

All further communications concerning your paper should quote reference number EMBOR-2020-52032V3 and be addressed to emboreports@wiley.com.

Should you be planning a Press Release on your article, please get in contact with emboreports@wiley.com as early as possible, in order to coordinate publication and release dates.

Corresponding Author Name: Stefano Santaguida

Manuscript Number: EMBOR-2020-52032-T